# Multimodal Dataset Distillation Made Simple by Prototype-Guided Data Synthesis

**Junhyeok Choi, Sangwoo Mo & Minwoo Chae***
Department of Industrial and Management Engineering
Pohang University of Science and Technology
{cjunh4810, sangwoo.mo, mchae}@postech.ac.kr

## Abstract

Recent advances in multimodal learning have achieved remarkable success across diverse vision–language tasks. However, such progress heavily relies on large-scale image–text datasets, making training costly and inefficient. Prior efforts in dataset filtering and pruning attempt to mitigate this issue, but still require relatively large subsets to maintain performance and fail under very small subsets. Dataset distillation offers a promising alternative, yet existing multimodal dataset distillation methods require full-dataset training and joint optimization of image pixels and text features, making them architecture-dependent and limiting cross-architecture generalization. To overcome this, we propose a learning-free dataset distillation framework that eliminates the need for large-scale training and optimization while enhancing generalization across architectures. Our method uses CLIP to extract aligned image–text embeddings, obtains prototypes, and employs an unCLIP decoder to synthesize images, enabling efficient and scalable multimodal dataset distillation. Extensive experiments demonstrate that our approach consistently outperforms optimization-based dataset distillation and subset selection methods, achieving state-of-the-art cross-architecture generalization. Our code is available at https://github.com/junhyeok9712/PDS.

## 1 Introduction

Multimodal models, such as CLIP (Radford et al., 2021), have emerged as a central paradigm in machine learning, demonstrating remarkable generalization across diverse tasks, including zero-shot classification and image–text retrieval. At the core of these successes are large-scale image–text datasets, such as LAION-5B (Schuhmann et al., 2022), whose use entails substantial computational and memory costs during training. This challenge naturally raises a critical question: *Can we substantially reduce the number of training samples without severely compromising performance?* Addressing this question is not only crucial for resource efficiency but also offers several practical advantages. Compact distilled datasets enable faster training, which facilitates rapid benchmarking of models and training strategies, including hyperparameter tuning and neural architecture search (Zhao et al., 2021). They are also effective in continual learning settings (Zhao & Bilen, 2021) that require quick adaptation to new data or tasks. Moreover, distilling massive datasets into smaller essential forms provides new insights into the core information required for effective learning.

A variety of approaches have been proposed to reduce dataset size, including coreset selection (Farahani & Hekmatfar, 2009; Welling, 2009), dataset filtering (Gadre et al., 2023), and pruning methods (Yang et al., 2023; Mahmoud et al., 2024), which have shown competitive performance in the context of CLIP. However, since they rely on representative subsets of the original dataset, they are effective only with sufficiently large reduced datasets and fail once the subset is too small to preserve semantic diversity. In contrast, dataset distillation (Wang et al., 2018) synthesizes new samples that encode the semantics of the original dataset, producing compact yet highly informative datasets. This makes it particularly effective on extremely reduced datasets, where it has outperformed selection-based strategies in image classification (Zhao & Bilen, 2021; Cazenavette et al., 2022), even with only a few distilled samples per class.

---

*Corresponding author

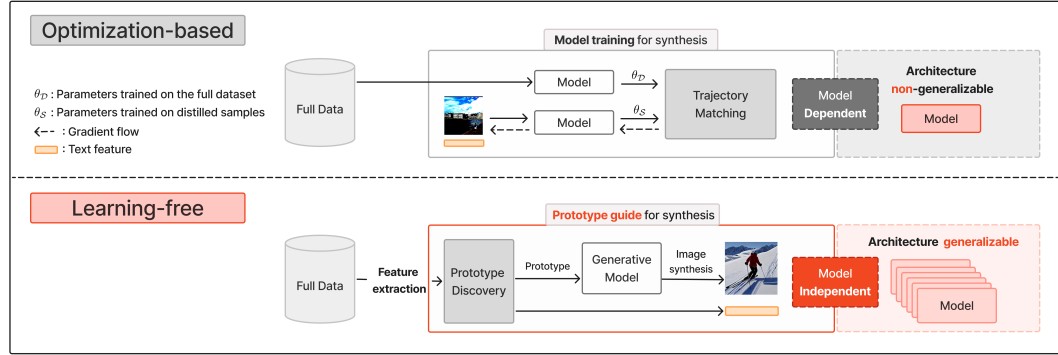

Figure 1: **Concept.** Optimization-based methods are computationally expensive and architecture-dependent. Our learning-free framework is simple, efficient, and generalizable across architectures.

However, research on multimodal dataset distillation remains underexplored, despite the increasing importance of multimodal learning. Existing approaches (Wu et al., 2024; Xu et al., 2024) are typically optimization-based, jointly optimizing image pixels and text features. Unfortunately, these methods face severe computational and scalability challenges. They require repeatedly training models on the full dataset, storing all intermediate parameters, and reusing them during sample synthesis. As datasets and model capacities grow, this process becomes prohibitively time-consuming and memory-intensive. Moreover, the number of parameters to optimize increases linearly with the number of synthetic samples, and numerous hyperparameters (e.g., learning rates for image and text) must be carefully tuned. More critically, these methods are architecture-dependent (Zhao et al., 2023b; Zhong & Liu, 2023). The resulting synthetic dataset is nearly identical to the original data, as the distillation process essentially adds architecture-dependent adversarial perturbations to the initialization images (see Figure 3). Consequently, these distilled datasets generalize poorly when applied to different model architectures, requiring the distillation process to be repeated from scratch. This requirement imposes substantial computational cost and severely limits the practical applicability of existing methods.

In this paper, we propose the prototype-guided data synthesis (PDS) framework, the first learning-free multimodal dataset distillation that is both simple and effective. Unlike optimization-based approaches, PDS requires no training or fine-tuning and avoids learning pixels and text features, making it computationally efficient, architecture-independent, and broadly generalizable (See Figure 1). Recent works have introduced learning-free methods for image classification (Su et al., 2024; Chan-Santiago et al., 2025), where images are generated for each class using explicit class labels. However, these methods cannot be extended to multimodal datasets because they do not consider cross-modal alignment. Aligned image–text pairs are essential for multimodal learning (Chen et al., 2020; Li et al., 2021). However, the VAE encoder (Kingma & Welling, 2014) commonly used in image-only distillation, produces embeddings that are not aligned with text embeddings from separate text encoders. Consequently, naively extending existing learning-free techniques produces synthetic datasets lacking semantic alignment across modalities. This motivates a new learning-free approach that enforces cross-modal alignment to synthesize semantically aligned image–text pairs.

We use CLIP (Radford et al., 2021) to capture image–text alignment in a learning-free manner. For each modality, we extract features and perform clustering to obtain semantically diverse clusters. We then formulate a linear assignment problem to align these clusters across modalities, obtaining image–text prototypes. These prototypes are then used to synthesize images. However, since standard Stable Diffusion models (Rombach et al., 2022; Podell et al., 2024) cannot generate images conditioned on CLIP image embeddings, we build on the idea of unCLIP (Ramesh et al., 2022; Patel et al., 2024), which enables such conditioning. Our ablation studies validate this design choice by confirming the importance of image prototype-based synthesis. Furthermore, experiments show that PDS generalizes well across architectures, reducing dependence on specific architectures and outperforming optimization-based multimodal dataset distillation methods. We also demonstrate the limitations of subset selection when the reduced dataset is too small, and show that image-only learning-free approaches cannot be extended to multimodal settings without cross-modal alignment.

## 2 RELATED WORK

**Dataset distillation.** Dataset distillation (Wang et al., 2018) seeks to synthesize a compact synthetic dataset that can replace a large-scale dataset while preserving performance close to training on the full dataset. Most prior work has primarily focused on image classification, aiming to synthesize a few representative images per class, with its potential applications in continual learning (Zhao & Bilen, 2021; Zhao et al., 2021; Zhao & Bilen, 2023), privacy-preserving data sharing (Li et al., 2020; Chen et al., 2022; Dong et al., 2022), and neural architecture search (Zhao et al., 2021; Zhao & Bilen, 2023). We first review optimization-based and generative approaches in image-only settings and then address multimodal dataset distillation.

**Optimization-based methods.** Early work formulates dataset distillation as a bi-level optimization problem, where synthetic data are optimized to match the training dynamics induced by the full dataset. These approaches generally incur substantial computational overhead, limiting their scalability to high-resolution images. Gradient matching (Zhao & Bilen, 2021; Zhao et al., 2021; Lee et al., 2022) enforces matching at the gradient level, whereas trajectory matching (Cazenavette et al., 2022; Cui et al., 2023; Du et al., 2023) matches the parameter trajectory during training to capture long-term dynamics. TESLA (Cui et al., 2023) makes trajectory matching more practical by proposing a variant that is more efficient in memory and computation time.

To further improve scalability, recent work has proposed several extensions. Decoupled optimization methods such as $SRe^2L$ (Yin et al., 2023) adopt a squeeze-recover-relabel procedure. Building on this, subsequent work explores multi-backbone generalization (Shao et al., 2024a), patch-based recovery (Sun et al., 2024), which was later extended with refinement strategies (Tran et al., 2025), and adaptive or category-aware optimization (Shao et al., 2024b). A complementary line of research parameterizes the synthetic dataset with compact factorized representations, such as shared bases with learnable recombination functions (Liu et al., 2022; Deng & Russakovsky, 2022; Wei et al., 2023) and feature-sharing mechanisms (Zheng et al., 2024; Zhang et al., 2025a). Alternative parameterizations include frequency-domain (Shin et al., 2023) and neural-field representations (Shin et al., 2025). In parallel, feature-distribution matching has also been extensively studied (Wang et al., 2022; Liu et al., 2023; Zhao & Bilen, 2023; Zhao et al., 2023a; Zhang et al., 2024).

**Generative approaches.** Beyond pixel-space optimization, generative models have emerged as an alternative paradigm. One line of work optimizes the latent codes of pretrained generators, including GANs (Zhao & Bilen, 2022; Zhang et al., 2023), StyleGAN variants (Cazenavette et al., 2023; Zhong et al., 2025), and latent diffusion models (Moser et al., 2024; Wang et al., 2025). Another line fine-tunes pretrained generators, such as class-conditional GANs (Zhang et al., 2023) or diffusion models (Gu et al., 2024; Su et al., 2024; Zou et al., 2025). While effective, these methods are still computationally expensive due to the large number of trainable parameters and repeated forward–backward passes. Recent work proposes learning-free approaches that leverage pretrained diffusion models without additional fine-tuning. Specifically, image embeddings are clustered within each class, and representative embeddings from each cluster are used to synthesize new images (Su et al., 2024; Chan-Santiago et al., 2025; Zhao et al., 2025). These methods achieve competitive performance while substantially reducing computational cost.

**Multimodal dataset distillation.** Existing multimodal dataset distillation approaches are optimization-based, while learning-free methods remain unexplored. The first work (Wu et al., 2024) adopted trajectory matching (Cazenavette et al., 2022) via bi-level optimization to jointly learn image pixels and text features. To mitigate the substantial time and memory costs, the TESLA framework (Cui et al., 2023) was later applied, and Xu et al. (2024) further extended this line of work by replacing the InfoNCE loss with a weighted binary cross-entropy loss and introducing a learnable similarity matrix between synthesized images and texts. Subsequent studies (Zhang et al., 2025b; Dang et al., 2025) refined these strategies, further advancing the optimization-based paradigm.

## 3 PROTOTYPE-GUIDED DATA SYNTHESIS

### 3.1 PROBLEM FORMULATION

Let a large-scale image–text dataset $\mathcal{D} = \{(x_n, y_n)\}_{n=1}^N$ be given, where $x_n$ denotes an image and $y_n$ its paired caption. The objective of multimodal dataset distillation is to synthesize a compact yet

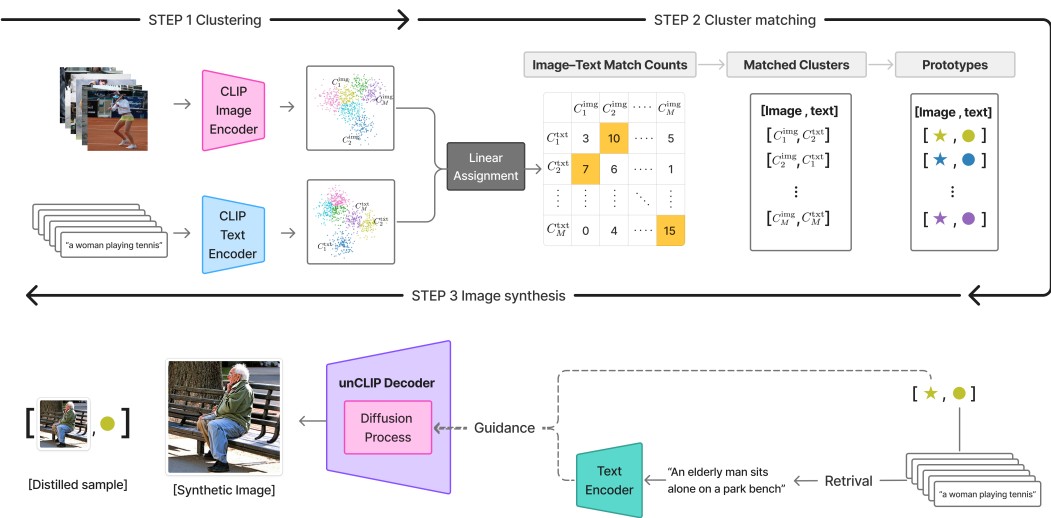

Figure 2: **Our PDS framework** distills multimodal datasets by synthesizing samples from image-text prototypes through three stages: (i) modality-specific clustering of CLIP embeddings, (ii) cross-modal cluster matching via a linear assignment to obtain prototypes, and (iii) image synthesis using an unCLIP decoder guided by image prototype and caption embeddings.

informative dataset $\mathcal{S} = \{(\tilde{x}_m, \tilde{y}_m)\}_{m=1}^M$, $M \ll N$, such that a vision-language model trained on $\mathcal{S}$ achieves performance without significant degradation compared to training on $\mathcal{D}$. This enables efficient model training with reduced computational and storage costs. Formally, for samples $\mathcal{B} = \{(x_i, y_i)\}_{i=1}^b$ from the underlying data distribution $\mathcal{P}$, the objective can be formulated as:

$$\mathcal{S}^* = \arg\min_{\mathcal{S}} \; \mathbb{E}_{\mathcal{B}\sim\mathcal{P}^{\otimes b}}\left[\left|\mathcal{L}\big(\mathcal{B};\boldsymbol{\theta}_{\mathcal{S}}\big) - \mathcal{L}\big(\mathcal{B};\boldsymbol{\theta}_{\mathcal{D}}\big)\right|\right] \quad \text{subject to} \quad \boldsymbol{\theta}_{\mathcal{X}} = \arg\min_{\boldsymbol{\theta}} \mathcal{L}(\mathcal{X};\boldsymbol{\theta}),$$

where $\mathcal{L}$ denotes a contrastive loss such as InfoNCE. In practice, since text is inherently discrete, dataset distillation is performed in the text embedding space, while images are parameterized in the pixel space. Existing methods typically rely on optimization-based approaches, which require repeated training on the full dataset and storage of intermediate parameters, resulting in significant computational and storage overhead. Moreover, their performance is architecture-dependent, often degrading when applied to models with different backbones.

## 3.2 OUR PDS FRAMEWORK

We propose a simple yet effective PDS framework for multimodal dataset distillation that uses pre-trained models without fine-tuning. The core idea of PDS is to identify image-text prototypes and synthesize new images based on these prototypes to construct a compact dataset. As shown in Figure 2, the pipeline consists of three steps: (i) clustering the CLIP embeddings of images and texts, (ii) matching the clusters to obtain cross-modal prototypes, and (iii) synthesizing images.

**Modality-specific clustering.** We adopt CLIP encoders (Radford et al., 2021) to obtain image and text embeddings that preserve high-level semantics and remain aligned across modalities. This alignment is particularly important for learning-free dataset distillation, where modality alignment cannot be learned. In fact, using an image encoder that is not aligned with a text encoder typically leads to poor performance, as shown by the VAE-based baselines in Table 3, thereby motivating the use of CLIP for both modalities. After extracting image and text embeddings $\{(z_n^{\text{img}}, z_n^{\text{txt}})\}_{n=1}^N$, we prune pairs with low similarity scores to remove noisy or weakly aligned pairs, resulting in a refined subset. We then perform clustering separately for each modality, producing clusters $\{C_m^{\text{img}}\}_{m=1}^M$ and $\{C_m^{\text{txt}}\}_{m=1}^M$ that group semantically similar embeddings and collectively capture the dataset's broad semantic diversity, where $M$ is set to the desired number of distilled samples. To handle large datasets efficiently, we adopt the mini-batch k-means algorithm (Hartigan & Wong, 1979; Sculley, 2010) (see Appendix C.1).

**Cluster matching for prototype construction.** While we obtain semantically coherent clusters within each modality, semantically related image–text clusters are not aligned across modalities. To

establish cross-modal correspondences, we measure the association between $C_i^{\text{img}}$ and $C_j^{\text{txt}}$ by the number of shared image-text pairs whose embeddings belong to the respective clusters. Clusters with more shared pairs are regarded as more semantically related, and we therefore aim to find a one-to-one cluster matching that maximizes the total number of shared pairs.

To this end, we formulate the problem as a linear assignment problem (Burkard & Cela, 1999), a framework widely used in logistics and scheduling. Specifically, we construct a cost matrix $K \in \mathbb{R}^{M \times M}$, where each entry represents the negative of the number of shared image-text pairs between a given image cluster and a text cluster:

$$K_{ij} = -\left| \left\{ (x_n, y_n) \mid z_n^{\text{img}} \in C_i^{\text{img}}, z_n^{\text{txt}} \in C_j^{\text{txt}} \right\} \right|.$$

Maximizing the total number of shared pairs is therefore equivalent to minimizing the total cost:

$$\min_{P \in \{0,1\}^{M \times M}} \sum_{i=1}^{M} \sum_{j=1}^{M} K_{ij} P_{ij} \quad \text{subject to} \quad \sum_{j=1}^{M} P_{ij} = 1, \ \sum_{i=1}^{M} P_{ij} = 1, \ P_{ij} \in \{0,1\},$$

where $P$ is a permutation matrix indicating the chosen matches. We solve this problem using the Hungarian algorithm (Munkres, 1957; Jonker & Volgenant, 1987), which computes the optimal one-to-one matching between image and text clusters.

For each matched cluster pair $(C_i^{\text{img}}, C_j^{\text{txt}})$, we retain only the embeddings of the shared image–text pairs. Embeddings whose paired embedding does not belong to the matched cluster of the other modality are discarded, as they are unlikely to be semantically related to the features in the matched cluster. This filtering strategy strengthens cross-modal alignment between matched clusters (See Appendix C.2). We then obtain the prototypes $(\tilde{z}_i^{\text{img}}, \tilde{z}_j^{\text{txt}})$ for each matched cluster pair by averaging the retained features. If no shared pairs exist, we instead use the original cluster centers to preserve the intended distilled dataset size. Matched clusters without shared pairs (hereafter simply referred to as pairless clusters) rarely appear when the distilled set size is small, and keeping or discarding them yields nearly identical performance. However, as the distilled set size increases, these pairless clusters become more frequent and their centroids tend to become misaligned, weakening cross-modal alignment and degrading performance. In these larger-scale regimes, discarding pairless clusters is therefore preferable. Additional discussion is provided in Appendix C.3. The resulting text prototypes are used as the distilled text features.

**Image synthesis.** We aim to synthesize images that encode information captured by image prototypes, since selecting real-image subsets does not adequately preserve the dataset's semantic diversity, which leads to lower performance as shown in Table 2. To synthesize images efficiently, we use a generative model rather than pixel-space optimization, as Table 4 demonstrates that direct optimization is slow and memory-intensive. However, existing learning-free dataset distillation methods cannot leverage image prototypes, as standard Stable Diffusion models (Rombach et al., 2022; Podell et al., 2024) do not condition their denoising U-Net (Ronneberger et al., 2015; Ho et al., 2020) on CLIP image embeddings. Similarly, text-to-image generation methods based on retrieved captions rely solely on text, leaving image prototypes unused.

To address this challenge, we build on the idea of unCLIP (Ramesh et al., 2022; Razzhigaev et al., 2023; Patel et al., 2024), which adopts a two-stage pipeline: a separately trained *prior* maps text into the CLIP image embedding space, and a *decoder* then generates images from these embeddings. In our approach, we employ only the unCLIP decoder, conditioning directly on CLIP image embeddings. This design enables the model to utilize image prototypes for generation, resulting in semantically rich synthesized images, as illustrated in Figure 3. In addition, it improves retrieval performance over baselines that either do not use image prototypes or use them only to select real images, as demonstrated in Table 5. We further strengthen semantic alignment by incorporating a text prototype. Because the unCLIP decoder cannot condition on CLIP text embeddings, we instead retrieve the caption most similar to the text prototype from the training set and use it as an additional condition. During diffusion, generation is guided by both the image prototype and the retrieved caption, maintaining semantic alignment. This additional caption condition further improves performance (Appendix C.4). To the best of our knowledge, this is the first dataset distillation method that generates images directly from CLIP image embeddings using an unCLIP decoder.

Table 1: **PDS distills datasets with superior cross-architecture generalization** compared to multimodal dataset distillation baselines. We evaluate on unseen vision backbones, comparing against TESLA-VL and LoRS on Flickr30K (top) and MS-COCO (bottom). Across all distilled dataset sizes and evaluation backbones, PDS consistently outperforms both baselines, demonstrating that its distilled datasets generalize effectively to unseen backbones.

| Evaluation Model | Pairs | Methods | IR@1 | IR@5 | IR@10 | TR@1 | TR@5 | TR@10 |
|---|---|---|---|---|---|---|---|---|
| ResNet | 100 | TESLA-VL | 4.1 ± 0.3 | 14.7 ± 0.9 | 22.9 ± 1.2 | 6.5 ± 0.4 | 17.8 ± 1.4 | 27.3 ± 1.4 |
| | | LoRS | 6.3 ± 0.1 | 18.6 ± 0.1 | 28.0 ± 0.2 | 9.1 ± 0.2 | 24.3 ± 0.4 | 34.5 ± 0.8 |
| | | PDS | **7.9 ± 0.3** | **25.8 ± 0.4** | **37.3 ± 0.3** | **10.2 ± 0.3** | **28.2 ± 0.9** | **39.0 ± 0.3** |
| | 300 | TESLA-VL | 10.3 ± 0.3 | 28.8 ± 0.6 | 40.6 ± 0.8 | 14.9 ± 0.9 | 36.2 ± 1.1 | 48.8 ± 1.7 |
| | | LoRS | 8.6 ± 0.9 | 23.5 ± 1.5 | 33.5 ± 1.8 | 14.7 ± 1.2 | 33.5 ± 2.8 | 44.1 ± 3.2 |
| | | PDS | **14.4 ± 0.4** | **38.1 ± 0.2** | **51.4 ± 0.4** | **18.7 ± 0.5** | **45.0 ± 0.4** | **57.8 ± 0.6** |
| | Full Dataset | | 28.5 ± 0.2 | 59.6 ± 0.1 | 71.4 ± 0.1 | 46.0 ± 0.6 | 76.2 ± 0.3 | 84.4 ± 0.2 |
| ViT | 100 | TESLA-VL | 2.1 ± 0.3 | 7.8 ± 0.7 | 13.1 ± 1.2 | 2.6 ± 0.6 | 8.7 ± 0.9 | 13.7 ± 1.4 |
| | | LoRS | 2.8 ± 0.1 | 9.9 ± 0.4 | 16.1 ± 0.2 | 5.2 ± 0.3 | 13.6 ± 0.5 | 20.5 ± 0.2 |
| | | PDS | **6.8 ± 0.3** | **19.2 ± 0.3** | **28.5 ± 0.4** | **6.6 ± 0.5** | **17.5 ± 0.5** | **26.9 ± 0.5** |
| | 300 | TESLA-VL | 5.1 ± 0.7 | 16.2 ± 1.0 | 24.5 ± 1.1 | 6.1 ± 0.6 | 18.0 ± 1.0 | 27.3 ± 1.4 |
| | | LoRS | 4.1 ± 0.5 | 13.1 ± 1.0 | 20.7 ± 1.3 | 6.2 ± 0.6 | 17.1 ± 1.2 | 25.7 ± 1.8 |
| | | PDS | **9.1 ± 0.1** | **27.3 ± 0.4** | **38.4 ± 0.4** | **9.6 ± 0.3** | **26.1 ± 0.5** | **37.5 ± 1.2** |
| | Full Dataset | | 22.7 ± 0.3 | 49.9 ± 0.3 | 62.3 ± 0.3 | 35.5 ± 0.4 | 62.9 ± 0.3 | 73.0 ± 0.4 |
| ResNet | 100 | TESLA-VL | 1.4 ± 0.1 | 5.8 ± 0.2 | 10.2 ± 0.4 | 2.1 ± 0.3 | 7.6 ± 0.3 | 12.5 ± 0.3 |
| | | LoRS | 1.8 ± 0.1 | 6.8 ± 0.2 | 11.4 ± 0.4 | 2.3 ± 0.2 | 7.8 ± 0.5 | 12.6 ± 0.4 |
| | | PDS | **2.8 ± 0.1** | **10.0 ± 0.2** | **17.3 ± 0.3** | **4.5 ± 0.2** | **14.0 ± 0.3** | **21.4 ± 0.4** |
| | 300 | TESLA-VL | 3.0 ± 0.1 | 10.7 ± 0.3 | 17.6 ± 0.4 | 5.7 ± 0.2 | 17.0 ± 0.3 | 25.6 ± 0.4 |
| | | LoRS | 2.5 ± 0.2 | 8.5 ± 0.4 | 13.8 ± 0.6 | 3.2 ± 0.2 | 10.1 ± 0.4 | 15.9 ± 0.6 |
| | | PDS | **5.3 ± 0.2** | **17.2 ± 0.4** | **27.2 ± 0.6** | **7.4 ± 0.3** | **20.7 ± 0.3** | **30.2 ± 0.4** |
| | Full Dataset | | 12.6 ± 0.1 | 33.4 ± 0.1 | 46.5 ± 0.2 | 26.6 ± 0.4 | 53.1 ± 0.2 | 66.1 ± 0.1 |
| ViT | 100 | TESLA-VL | 0.5 ± 0.2 | 2.1 ± 0.8 | 3.8 ± 1.4 | 0.2 ± 0.1 | 1.0 ± 0.5 | 1.7 ± 0.9 |
| | | LoRS | 0.8 ± 0.1 | 3.0 ± 0.2 | 5.4 ± 0.4 | 0.7 ± 0.2 | 2.4 ± 0.3 | 4.1 ± 0.5 |
| | | PDS | **2.3 ± 0.1** | **8.6 ± 0.1** | **14.5 ± 0.1** | **2.2 ± 0.1** | **7.7 ± 0.2** | **13.2 ± 0.3** |
| | 300 | TESLA-VL | 1.5 ± 0.1 | 5.9 ± 0.1 | 10.1 ± 0.3 | 2.2 ± 0.2 | 7.8 ± 0.2 | 12.6 ± 0.4 |
| | | LoRS | 1.0 ± 0.3 | 3.7 ± 1.1 | 6.3 ± 1.8 | 1.3 ± 0.4 | 4.6 ± 1.4 | 7.4 ± 2.0 |
| | | PDS | **4.1 ± 0.1** | **13.4 ± 0.1** | **21.2 ± 0.1** | **3.7 ± 0.1** | **12.6 ± 0.1** | **19.8 ± 0.2** |
| | Full Dataset | | 11.5 ± 0.1 | 29.8 ± 0.1 | 41.7 ± 0.1 | 19.5 ± 0.2 | 42.4 ± 0.4 | 55.3 ± 0.3 |

## 4 EXPERIMENTS

This section presents extensive empirical results that validate the effectiveness of PDS. Performance comparisons highlight its strengths in cross-generalization, effectiveness on extremely reduced datasets, and suitability for multimodal datasets. In addition, the ablation studies support our design choices, including the use of generative models and prototype-based image synthesis. In all tables, the best results are highlighted in **bold**, and the second-best are underlined.

### 4.1 EXPERIMENTAL SETTINGS

**Datasets and metrics.** We evaluate our approach on two widely used vision-language benchmarks: Flickr30K (Plummer et al., 2015) (29,000/1,000 images for train/test) and MS-COCO (Lin et al., 2014) (113,000/5,000 images for train/test), where each image is paired with five human-annotated captions. Model performance is evaluated using recall at $k$ (R@$k$) for cross-modal retrieval, with $k \in \{1, 5, 10\}$. For each query from one modality, we compute cosine similarity with all candidates from the other modality in the test set, retrieve the top-$k$ most similar samples, and check whether the ground-truth match is included among them. We report text-to-image retrieval as IR@$k$ and image-to-text retrieval as TR@$k$.

**Implementation details.** Unless otherwise specified, all baselines use CLIP ViT-L/14 (Radford et al., 2021). For PDS, images are generated with the unCLIP decoder using classifier-free guidance (Ho & Salimans, 2022) with a guidance scale of 5.0 and 100 sampling steps, and resized to $224 \times 224$ for evaluation. To assess cross-architecture generalization, we follow Xu et al. (2024) and evaluate whether the distilled datasets remain effective when transferred to alternative vision backbones such as ResNet-50 (He et al., 2016) and ViT-Ti/16 (Dosovitskiy et al., 2021). For this evaluation, we use each alternative backbone as the image encoder and attach a learnable linear projection layer after the text encoder. We then fine-tune the CLIP-style model on the distilled dataset while keeping the text encoder frozen. For all baselines, we adopt the hyperparameters reported in the original papers, and conduct all experiments on a single RTX 3090 GPU.

## 4.2 PERFORMANCE COMPARISON

We evaluate PDS against three categories of baselines: multimodal dataset distillation methods, dataset subset selection methods, and learning-free distillation methods for image classification. These comparisons are designed to highlight three aspects of our contribution: (i) mitigating the cross-generalization limitations of existing multimodal dataset distillation methods, (ii) demonstrating the advantage of PDS over subset selection when the reduced dataset is extremely small, and (iii) overcoming the limitations of existing learning-free distillation approaches when extending from image classification to multimodal datasets. Detailed descriptions of each baseline are provided in Appendix A.

**Multimodal dataset distillation.** A key limitation of architecture-dependent distilled datasets is that they require re-distillation whenever a new backbone is introduced. To examine whether the distilled dataset can generalize to unseen backbones, we compare PDS against multimodal dataset distillation baselines, TESLA-VL and LoRS (Xu et al., 2024), in a cross-architecture setting. We exclude MTT-VL (Wu et al., 2024) due to its higher computational and memory cost as well as its lower performance compared to these methods. The baselines use pretrained NFNet (Brock et al., 2021) as image encoder, following Xu et al. (2024), while we replace the original BERT text encoder (Devlin et al., 2019) with the CLIP text encoder for a fair comparison. This replacement also improves performance (see Appendix C.5).

As shown in Table 1, PDS consistently outperforms both baselines on Flickr30K (Plummer et al., 2015) and MS-COCO (Lin et al., 2014) across all distilled pair sizes and evaluation backbones. For example, with 300 distilled pairs and a ResNet backbone, PDS achieves 14.4% IR@1 and 18.7% TR@1 on Flickr30K, surpassing baselines over 4.1 and 3.8 percentage points (pp), respectively. The performance gap increases at higher retrieval cutoffs (e.g., +10.8 pp in IR@10). These results reveal a key limitation of optimization-based baselines. Their distilled datasets are strongly architecture-dependent and require expensive re-distillation for each new backbone. In contrast, PDS produces distilled datasets that generalize well across architectures through a simple and efficient distillation process, enabling broad practical applicability.

**Dataset subset selection.** To obtain a highly reduced yet informative dataset, representative images must preserve broad semantic coverage. In this case, dataset distillation offers a more effective solution than subset selection, as it synthesizes data to preserve diverse semantic features. To validate this, we evaluate PDS against classical coreset-selection methods, K-center (Farahani & Hekmatfar, 2009) and Herding (Welling, 2009), as well as widely used filtering schemes such as CLIP score filtering, LAION filtering, and image-based filtering (Gadre et al., 2023).

As shown in Table 2, PDS consistently achieves the highest performance, outperforming the strongest baseline (Herding) by 17.2 and 10.8 pp in IR@10 and TR@10, respectively. Although dataset filtering methods are widely used in CLIP training due to their effectiveness with moderately reduced datasets, they fail with extremely reduced datasets. These results demonstrate that PDS is particularly well-suited for such scenarios, as it preserves broader semantic coverage by interpolating diverse semantic features and achieves superior performance.

**Learning-free approaches for image classification.** We examine whether learning-free dataset distillation methods for image classification can be extended to the multimodal setting. To investigate this, we extend $D^4M$ (Su et al., 2024) and $MGD^3$ (Chan-Santiago et al., 2025) by retaining their Stable Diffusion VAE (Rombach et al., 2022) as the image encoder and additionally incorpo-

Table 2: **The representative samples obtained by interpolating diverse semantic features are more suitable than subsets of the original dataset when the reduced dataset is extremely small.** We compare PDS with subset selection methods in the 100-pair setting on Flickr30K. Baselines restricted to real samples are limited in preserving broad semantic diversity. In contrast, PDS synthesizes representative samples that reflect diverse semantics and consistently achieves superior results.

| Evaluation Model | Metric | Subset selection | | | | | Distillation |
|---|---|---|---|---|---|---|---|
| | | K-center | Herding | CLIP score | LAION filtering | Image-based | PDS |
| ResNet | IR@1 | 2.9 ± 0.1 | 3.6 ± 0.2 | 2.5 ± 0.1 | 2.4 ± 0.1 | 2.2 ± 0.2 | **7.9 ± 0.3** |
| | IR@5 | 10.4 ± 0.3 | 12.6 ± 0.5 | 8.7 ± 0.3 | 8.5 ± 0.2 | 8.6 ± 0.3 | **25.8 ± 0.4** |
| | IR@10 | 16.8 ± 0.3 | 20.1 ± 0.6 | 14.5 ± 0.3 | 14.5 ± 0.3 | 13.6 ± 0.4 | **37.3 ± 0.3** |
| | TR@1 | 5.3 ± 0.3 | 6.7 ± 0.3 | 4.7 ± 0.4 | 4.7 ± 0.3 | 4.0 ± 0.3 | **10.2 ± 0.3** |
| | TR@5 | 15.6 ± 0.5 | 19.4 ± 0.7 | 12.6 ± 0.7 | 12.4 ± 0.6 | 10.5 ± 0.4 | **28.2 ± 0.9** |
| | TR@10 | 24.1 ± 0.4 | 28.2 ± 0.6 | 18.9 ± 0.7 | 19.3 ± 0.4 | 16.2 ± 0.6 | **39.0 ± 0.3** |

Table 3: **Cross-modal alignment is crucial for effective multimodal dataset distillation.** We compare PDS with learning-free methods for image classification in the 100-pair setting on Flickr30K. Both $D^4M$ and $MGD^3$ rely on the VAE encoder, which produces image features that are not aligned with text features, leading to low retrieval accuracy. In contrast, PDS uses CLIP encoders to enforce cross-modal alignment, outperforming the baselines and highlighting the importance of alignment-aware distillation.

| Evaluation Model | Methods | IR@1 | IR@5 | IR@10 | TR@1 | TR@5 | TR@10 |
|---|---|---|---|---|---|---|---|
| ResNet | $D^4M$ | 1.3 ± 0.1 | 5.8 ± 0.2 | 9.8 ± 0.3 | 1.4 ± 0.2 | 5.6 ± 0.5 | 9.5 ± 0.3 |
| | $MGD^3$ | 2.6 ± 0.1 | 10.0 ± 0.2 | 17.2 ± 0.5 | 3.4 ± 0.2 | 11.6 ± 0.3 | 17.5 ± 0.4 |
| | PDS | **7.9 ± 0.3** | **25.8 ± 0.4** | **37.3 ± 0.3** | **10.2 ± 0.3** | **28.2 ± 0.9** | **39.0 ± 0.3** |

rating a CLIP text encoder. As in our method, prototypes are obtained through clustering and cluster matching, after which each baseline applies its original procedure to synthesize images.

Although these methods perform competitively on image classification, their reliance on the VAE encoder leads to poor alignment with CLIP text embeddings and consequently degrades retrieval performance (Table 3). For example, with ResNet, PDS achieves 37.3% IR@10 and 39.0% TR@10, whereas $MGD^3$ reaches only 17.2% and 17.5%. These findings highlight the importance of strong cross-modal alignment for effective multimodal dataset distillation and demonstrate the limitations of existing learning-free methods in multimodal settings.

## 4.3 ABLATION STUDIES

**Advantage of using a generative model.** We analyze the benefits of using a generative model by comparing PDS with optimization-based multimodal dataset distillation methods. Such optimization-based approaches often synthesize images nearly identical to the initialization sample (Figure 3, left), essentially adding architecture-dependent adversarial perturbations rather than generating genuinely new data, thereby limiting semantic diversity. In contrast, PDS employs a generative model to synthesize images from prototypes constructed by interpolating diverse semantic features.

PDS further improves efficiency by eliminating pixel-space optimization. To validate this, we compare PDS with CLIP inversion (Kazemi et al., 2024), which directly optimizes pixel values so that the CLIP embedding of a synthesized image aligns with either an image or text prototype. We evaluate two variants: (i) Image prototype alignment and (ii) Text prototype alignment. As shown in Table 4, PDS reduces peak memory usage (from 6.13 GB to 4.34 GB) and dramatically shortens the generation time (from 1,477 s to 9.7 s per image). In addition, CLIP inversion produces unrealistic images (Figure 3, middle), resulting in poor performance (Table 4).

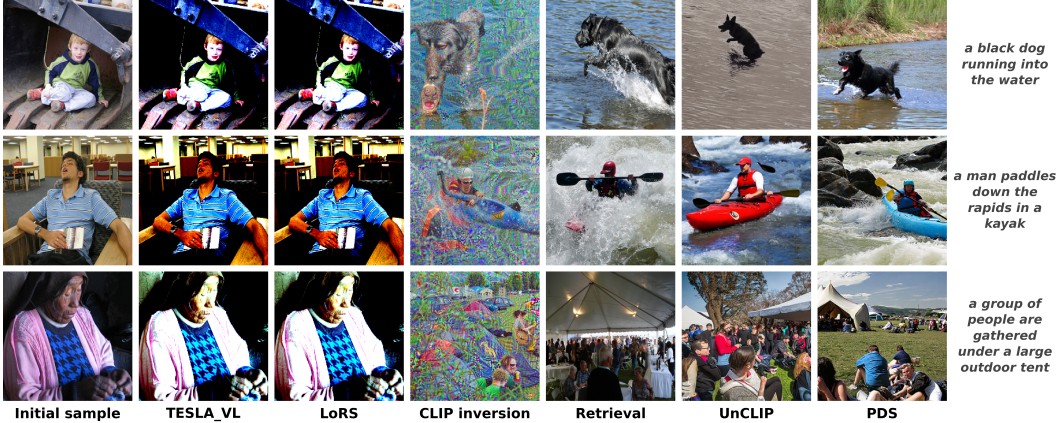

Figure 3: **The synthesized images.** Left (col. 1-3): Images from multimodal dataset distillation baselines, which are nearly identical to the initialization image. Middle: Image generated via CLIP inversion from image prototypes, which is not realistic. Right (col. 5-7): Given a text prototype, we first retrieve the most similar caption and present three images: the paired real image, the UnCLIP-generated image from this caption, and the PDS-generated image. Unlike UnCLIP, which strictly follows captions, PDS produces realistic and semantically enriched images by conditioning on image prototypes.

Table 4: **Using a generative model for image synthesis improves both efficiency and performance.** We evaluate two variants of CLIP inversion: (i) Image prototype alignment and (ii) Text prototype alignment, both of which directly optimize pixel values, in the 100-pair setting on Flickr30K. By employing a generative model, PDS accelerates image generation, reduces memory usage, and produces realistic images, achieving superior performance over CLIP inversion.

| Evaluation Model | Methods | IR@1 | TR@1 | Memory (GB) | Time (s) |
|---|---|---|---|---|---|
| ResNet | Text alignment | $1.4 \pm 0.1$ | $2.0 \pm 0.5$ | 6.13 | 1477.71 |
| | Image alignment | $\underline{4.4 \pm 0.2}$ | $\underline{4.2 \pm 0.3}$ | | |
| | PDS | $\mathbf{7.9 \pm 0.3}$ | $\mathbf{10.2 \pm 0.3}$ | **4.34** | **9.71** |

Table 5: **Incorporating an image prototype through the unCLIP decoder improves performance.** We compare PDS with two retrieval baselines and an unCLIP-based text-to-image generation method in the 100-pair setting on Flickr30K. While the retrieval baselines rely on real images and unCLIP generates images from text alone, PDS incorporates image prototypes and textual information through the unCLIP decoder, leading to superior performance.

| Evaluation Model | Methods | IR@1 | IR@5 | IR@10 | TR@1 | TR@5 | TR@10 |
|---|---|---|---|---|---|---|---|
| ResNet | Text-prototype | $5.2 \pm 0.1$ | $17.5 \pm 0.2$ | $27.1 \pm 0.1$ | $6.4 \pm 0.2$ | $19.4 \pm 0.4$ | $28.2 \pm 0.6$ |
| | Image-prototype | $\underline{5.5 \pm 0.3}$ | $\underline{19.0 \pm 0.1}$ | $\underline{28.7 \pm 0.4}$ | $\underline{8.0 \pm 0.4}$ | $\underline{20.4 \pm 0.6}$ | $\underline{30.2 \pm 0.6}$ |
| | unCLIP | $5.2 \pm 0.2$ | $17.1 \pm 0.4$ | $26.7 \pm 0.3$ | $6.4 \pm 0.5$ | $19.5 \pm 0.8$ | $28.9 \pm 0.6$ |
| | PDS | $\mathbf{7.9 \pm 0.3}$ | $\mathbf{25.8 \pm 0.4}$ | $\mathbf{37.3 \pm 0.3}$ | $\mathbf{10.2 \pm 0.3}$ | $\mathbf{28.2 \pm 0.9}$ | $\mathbf{39.0 \pm 0.3}$ |

**Impact of image prototype-based image generation.** To assess the impact of image prototype-based generation, we compare PDS with three baselines that obtain images through different mechanisms. First, we consider two baselines that do not use image prototypes. The first is text prototype-based retrieval, which retrieves the caption in the training set whose embedding has the highest cosine similarity to the given text prototype and then selects its paired real image. The second is an unCLIP-based text-to-image generation method (Ramesh et al., 2022), which applies the full un-CLIP pipeline (prior and decoder) to generate an image from the retrieved caption. In addition, we consider image prototype-based retrieval, which uses the image prototype only to retrieve the most similar real image from the training set without using it for generation.

Figure 3 (right) illustrates the benefits of incorporating an image prototype into image generation. PDS generates images with richer visual details, producing backgrounds that not only align with the caption but also reflect the interpolated semantic information encoded in the image prototype, resulting in scenes contextually consistent with both. Quantitatively, PDS consistently outperforms baselines that do not use image prototypes (Table 5). For example, with ResNet, IR@10 improves from 26.7% (unCLIP) and 27.1% (Text-prototype) to 37.3%, with similar gains observed for TR@10. The performance remains substantially below that of PDS even when using the image prototype to retrieve real images (Image-prototype). These results demonstrate that using an image prototype for image generation is crucial for producing compact yet highly informative synthetic samples and achieving strong retrieval performance beyond what can be obtained from captions alone or from retrieving real images.

Additional ablation studies and extended analyses are provided in Appendix C, including comprehensive analyses of PDS-related hyperparameters, distilled dataset sizes, and alternative decoder conditioning strategies. This appendix also examines robustness on rare samples, evaluates generalization to a different CLIP variant, and reports results on an additional dataset. Appendix D presents two applications of our method, demonstrating how the distilled dataset can be used within ASIF (Norelli et al., 2023) and how PDS can be incorporated into an optimization-based distillation framework.

## 5 CONCLUSION

**Summary.** In this work, we introduce PDS, a learning-free approach to multimodal dataset distillation that considers image–text alignment and synthesizes images using image–text prototypes. PDS is scalable and efficient, achieving strong cross-generalization and maintaining significant advantages even with an extremely reduced dataset. This work offers a new perspective on multimodal dataset distillation and demonstrates a more effective and generalizable strategy.

**Limitations.** PDS leverages a pre-trained generative model capable of synthesizing images conditioned on embeddings. Although stronger models such as SigLIP (Zhai et al., 2023) provide better image–text alignment, the absence of generative models that can be conditioned on these embeddings prevents their direct use for learning-free dataset distillation, suggesting a promising direction for future work. Additionally, the prototypes may be more influenced by dominant classes corresponding to frequently occurring concepts, while rare or long-tail classes may be underrepresented. However, this limitation is shared by subset selection and other distillation baselines, and PDS shows stronger robustness on rare samples than these methods (see Appendix C.9). A separate challenge arises when distilling datasets from domains that differ significantly from those used to train CLIP and the unCLIP decoder. Since these models are trained primarily on natural images, they perform poorly in specialized areas such as medical imaging, and image generation conditioned on such embeddings may fail. In such cases, both CLIP and the unCLIP decoder need to be fine-tuned.

### ACKNOWLEDGMENTS

**Funding.** This work was supported by the National Research Foundation of Korea (NRF) grant funded by the Korea government (MSIT) (No. RS-2023-00240861), a Korea Institute for Advancement of Technology (KIAT) grant funded by the Korea government (MOTIE) (RS-2024-00409092, 2024 HRD Program for Industrial Innovation), Institute of Information & Communications Technology Planning & Evaluation (IITP)-Global Data-X Leader HRD program grant funded by the Korea government (MSIT) (IITP-2024-RS-2024-00441244), and Korea Planning & Evaluation Institute of Industrial Technology (KEIT) funded by the Ministry of Trade, Industry and Resources (No. RS-2025-25458052, Development of Core Technologies for Manufacturing Foundation Models)

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

LLM USAGE STATEMENT

We employed large language models solely for polishing the writing and checking grammar.

## A  BASELINES

**Dataset subset selection**

- **Herding** (Welling, 2009). This method selects samples so that the subset mean approximates the full-dataset mean in the feature space. Starting from an empty set, it iteratively adds the image-text pair whose inclusion minimizes the distance between the updated subset mean and the full-dataset mean.

- **K-center** (Farahani & Hekmatfar, 2009). This method selects samples to maximize dataset coverage by choosing samples that are maximally separated. At each step, it selects the sample whose minimum distance to the selected set is maximized.

- **CLIP score filtering**. This method selects image-text pairs whose CLIP-based cosine similarity exceeds a predefined threshold. In our experiments, we retain the top-k pairs from the candidate set.

- **LAION filtering**. This method follows the LAION-2B filtering scheme in two steps. It first applies cld3 English filtering, retaining pairs whose text is classified as English with probability above a predefined threshold. It then applies CLIP score filtering to select pairs with a high CLIP score.

- **Image-based filtering**. This method constructs a subset aligned with the ImageNet classes. It first applies fastText-based English filtering and caption-length filtering. CLIP image embeddings are extracted from the remaining dataset and clustered using FAISS (Johnson et al., 2019). CLIP embeddings of ImageNet images are then assigned to these clusters, and only clusters containing at least one ImageNet embedding are retained. Samples in these retained clusters form the subset, which is further refined using CLIP score filtering.

**Multimodal dataset distillation**

- **TESLA-VL** (Xu et al., 2024). This method applies the TESLA framework (Cui et al., 2023) to reduce the computational and memory costs of trajectory matching in the original work (Wu et al., 2024), and further replaces the InfoNCE loss with a weighted binary cross-entropy loss.

- **LoRS** (Xu et al., 2024). This method extends TESLA-VL by learning an additional similarity matrix between synthesized images and texts.

**Learning-free dataset distillation for image classification**

- **D$^4$M** (Su et al., 2024). This method leverages a latent diffusion model (LDM) and introduces an initialization strategy based on cluster prototypes. Features are extracted from the dataset using a VAE encoder, and k-means clustering is performed for each class to obtain class-specific prototypes. These prototypes generate noise through the forward diffusion process, and the resulting noise is used to initialize the reverse diffusion process.

- **MGD$^3$** (Chan-Santiago et al., 2025). This approach follows a strategy similar to D$^4$M. Features are extracted with a VAE encoder, and k-means clustering is applied within each class to obtain prototypes. These prototypes then serve as explicit guidance during the reverse diffusion process.

# B  ALGORITHM

---

**Algorithm 1 PDS framework**

---

**Require:** Original real dataset $\mathcal{D}$; CLIP image ans text encoder; unCLIP decoder.
1: ▷ *Modality-Specific Clustering*
2: Extract the CLIP image and text embeddings $(z^{\text{img}}, z^{\text{txt}})$.
3: Perform clustering separately on the image and text embeddings to obtain $C^{\text{img}}$ and $C^{\text{txt}}$.
4: ▷ *Cluster Matching for Prototype Construction*
5: Solve the linear assignment problem:

$$\min_{P \in \{0,1\}^{M \times M}} \sum_{i=1}^{M} \sum_{j=1}^{M} K_{ij} P_{ij} \text{ subject to } \sum_{j=1}^{M} P_{ij} = 1, \ \sum_{i=1}^{M} P_{ij} = 1, \ P_{ij} \in \{0,1\},$$

where $K_{ij}$ denotes the negative of the number of shared pairs between cluster $i$ and $j$.
6: For each matched cluster pair, obtain the prototypes $(\tilde{z}^{\text{img}}, \tilde{z}^{\text{txt}})$.
7: ▷ *Image Synthesis*
8: Synthesize images from the prototypes using the unCLIP decoder.
**Ensure:** Synthetic dataset $\mathcal{S}$

---

# C  ADDITIONAL ABLATION STUDIES

## C.1  EFFECT OF THE CLUSTERING ALGORITHM

We investigate how the choice of clustering algorithm affects overall performance by applying Gaussian mixture models (GMM; (Dempster et al., 1977)), agglomerative clustering (Ward Jr, 1963), and mini-batch k-means (Sculley, 2010). As shown in Table 6, PDS achieves robust performance across different clustering methods, indicating low sensitivity to the choice of clustering algorithm. Although GMM and agglomerative clustering achieve slightly better performance, their standard implementations require full-batch processing without mini-batch support, making them computationally prohibitive for large-scale data (see Table 7, measured on single-modality clustering). In contrast, mini-batch k-means scales efficiently and remains practical for large datasets, making it a suitable choice for real-world scenarios.

Table 6: **Comparison of clustering methods.** We evaluate PDS in the 100-pair setting on Flickr30K using different clustering algorithms. PDS achieves robust performance, indicating low sensitivity to the choice of clustering algorithm.

| Evaluation Model | Methods | IR@1 | IR@5 | IR@10 | TR@1 | TR@5 | TR@10 |
|---|---|---|---|---|---|---|---|
| ResNet | GMM | 8.3±0.1 | 26.2±0.3 | 37.4±0.2 | 10.9±0.4 | 30.1±0.4 | 41.1±0.6 |
| | Agglomerative | **9.3±0.2** | **28.9±0.3** | **41.4±0.1** | **11.6±0.6** | **30.2±0.3** | **41.6±0.6** |
| | Mini-batch k-means | 7.9±0.3 | 25.8±0.4 | 37.3±0.3 | 10.2±0.3 | 28.2±0.9 | 39.0±0.3 |
| ViT | GMM | 6.0±0.4 | 19.2±0.3 | 28.3±0.1 | 6.1±0.4 | 18.2±0.9 | 26.9±0.7 |
| | Agglomerative | 6.3±0.2 | **19.9±0.2** | **29.5±0.4** | 6.4±0.5 | **18.5±0.3** | **27.4±0.4** |
| | Mini-batch k-means | **6.8±0.3** | 19.2±0.3 | 28.5±0.4 | **6.6±0.5** | 17.5±0.5 | 26.9±0.5 |

Table 7: **Clustering time comparison.** Clustering time (in seconds) across different methods.

| | GMM | Agglomerative | Mini-batch k-means |
|---|---|---|---|
| Clustering Time (s) | 1,469.7 | 4,387.9 | **9.0** |

## C.2 Effect of the Filtering strategy

When obtaining image-text prototypes from the matched cluster pairs, we use only the embeddings of the shared image-text pairs. In this section, we compare this filtering strategy against a baseline that uses all embeddings without exclusion. Figure 4 presents histograms of cosine similarity between paired image and text prototypes under both settings. Filtering results in higher similarity, indicating stronger cross-modal alignment and ultimately leading to performance gains, as shown in Table 8. Applying the filtering strategy significantly improves performance over the non-filtering baseline. For example, with ResNet, IR@10 improves from 28.1% to 37.3% and TR@10 from 30.6% to 39.0%.

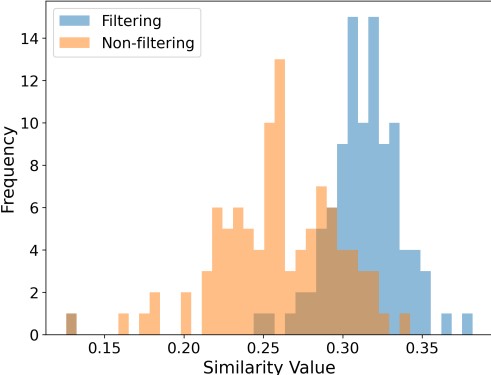

Figure 4: **Histograms of cosine similarity** between paired image and text prototypes in the 100-pair setting on Flickr30K, shown with and without filtering. Filtering leads to higher similarity.

Table 8: **Filtering enhances cross-modal alignment.** We compare our filtering strategy against a non-filtering baseline in the 100-pair setting on Flickr30K. Prototypes obtained from embeddings of the shared image–text pairs consistently improve performance.

| Evaluation Model | Methods | IR@1 | IR@5 | IR@10 | TR@1 | TR@5 | TR@10 |
|---|---|---|---|---|---|---|---|
| ResNet | Non-filtering | 5.3 ± 0.1 | 18.4 ± 0.1 | 28.1 ± 0.3 | 7.4 ± 0.5 | 21.2 ± 0.7 | 30.6 ± 0.5 |
| | Filtering | **7.9** ± 0.3 | **25.8** ± 0.4 | **37.3** ± 0.3 | **10.2** ± 0.3 | **28.2** ± 0.9 | **39.0** ± 0.3 |
| ViT | Non-filtering | 3.9 ± 0.0 | 14.0 ± 0.0 | 21.3 ± 0.0 | 4.5 ± 0.0 | 12.4 ± 0.0 | 19.6 ± 0.0 |
| | Filtering | **6.8** ± 0.3 | **19.2** ± 0.3 | **28.5** ± 0.4 | **6.6** ± 0.5 | **17.5** ± 0.5 | **26.9** ± 0.5 |

C.3 EFFECTS OF PAIRLESS CLUSTERS IN LARGE-SCALE DISTILLED DATASET

To more comprehensively evaluate the effect of matched clusters without shared pairs (pairless clusters), the two strategies (keeping all clusters or discarding the pairless clusters) are applied across a wide range of distilled dataset sizes. In the small-scale settings, both approaches show nearly identical performance because only a few pairless clusters arise. However, as the distilled size increases, the number of such clusters grows substantially, and the pairless clusters tend to have misaligned centroids (see Table 9 and Figure 5). This misalignment weakens cross-modal alignment and leads to the performance degradation reported in Table 10. At larger distilled sizes, discarding pairless clusters is therefore preferable.

Table 9: **Larger distilled sizes yield more pairless clusters.** We distill Flickr30K into various numbers of pairs. As the distilled size increases, the number of pairless clusters grows substantially.

|  | **Pairs** | | | | |
| --- | --- | --- | --- | --- | --- |
|  | 100 | 300 | 500 | 1000 | 1500 |
| **Number of pairless clusters** | 1 | 8 | 24 | 87 | 166 |

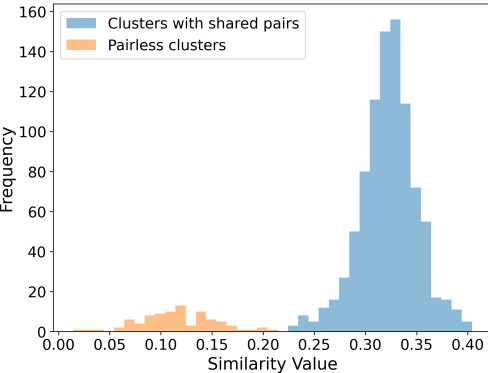

Figure 5: **Histograms of cosine similarity** between paired image and text prototypes in the 1000-pair setting on Flickr30K, shown for pairless clusters and clusters with shared pairs. Pairless clusters exhibit lower similarity.

Table 10: **Discarding pairless clusters becomes increasingly beneficial at larger distilled sizes.** We distill Flickr30K to a wide range of distilled dataset sizes and compare two strategies, keeping all clusters or discarding the pairless clusters. While both strategies perform similarly in small-scale settings, discarding pairless clusters becomes preferable when the distilled size becomes large.

| Evaluation Model | Pairs | Methods | IR@1 | IR@5 | IR@10 | TR@1 | TR@5 | TR@10 |
| --- | --- | --- | --- | --- | --- | --- | --- | --- |
| ResNet | 100 | Keeping | **7.9 ± 0.3** | 25.8 ± 0.4 | 37.3 ± 0.3 | **10.2 ± 0.3** | **28.2 ± 0.9** | **39.0 ± 0.3** |
|  |  | Discarding | 7.8 ± 0.2 | **25.9 ± 0.4** | **37.5 ± 0.2** | 10.0 ± 0.3 | 27.5 ± 1.2 | 38.8 ± 0.9 |
|  | 300 | Keeping | 14.4 ± 0.4 | 38.1 ± 0.2 | 51.4 ± 0.4 | 18.7 ± 0.5 | **45.0 ± 0.4** | **57.8 ± 0.6** |
|  |  | Discarding | **14.9 ± 0.2** | **38.6 ± 0.5** | **51.9 ± 0.3** | **19.3 ± 0.4** | **45.0 ± 0.5** | 57.2 ± 0.6 |
|  | 500 | Keeping | 16.3 ± 0.2 | 41.3 ± 0.2 | 54.9 ± 0.4 | 22.5 ± 0.3 | 49.3 ± 0.8 | 61.4 ± 0.4 |
|  |  | Discarding | **17.1 ± 0.2** | **43.0 ± 0.2** | **56.4 ± 0.1** | **23.9 ± 0.9** | **50.5 ± 0.6** | **62.0 ± 0.3** |
|  | 1000 | Keeping | 18.2 ± 0.3 | 43.5 ± 0.2 | 56.4 ± 0.5 | 24.3 ± 0.4 | 52.7 ± 0.6 | 64.7 ± 0.6 |
|  |  | Discarding | **19.5 ± 0.3** | **46.4 ± 0.4** | **58.9 ± 0.2** | **27.4 ± 0.4** | **55.3 ± 0.6** | **67.5 ± 0.6** |
|  | 1500 | Keeping | 18.9 ± 0.2 | 45.6 ± 0.2 | 59.4 ± 0.3 | 26.5 ± 0.3 | 55.5 ± 0.8 | 67.8 ± 1.2 |
|  |  | Discarding | **20.5 ± 0.2** | **48.7 ± 0.3** | **62.5 ± 0.2** | **28.8 ± 0.5** | **59.2 ± 0.8** | **71.2 ± 0.4** |

## C.4 EFFECT OF INCORPORATING CAPTIONS RETRIEVED FROM THE TEXT PROTOTYPE

We generate images conditioned not only on the image prototype but also on a caption retrieved from the text prototype, to enhance semantic alignment between the generated image and the text prototype. To evaluate the benefit of this design, we compare our method with a baseline that generates images conditioned solely on the image prototype. As shown in Table 11, incorporating the caption improves performance.

Table 11: **Joint conditioning improves performance.** We compare our method with a baseline that generates images conditioned solely on the image prototype in the 100-pair setting on Flickr30K. Conditioning on both the image prototype and a caption retrieved from the text prototype improves performance.

| Evaluation Model | Methods | IR@1 | IR@5 | IR@10 | TR@1 | TR@5 | TR@10 |
|---|---|---|---|---|---|---|---|
| ResNet | Image prototype | 6.0 ± 0.3 | 19.2 ± 0.3 | 28.3 ± 0.1 | 6.1 ± 0.4 | 18.2 ± 0.9 | 26.9 ± 0.7 |
| | Both prototypes | **7.9 ± 0.3** | **25.8 ± 0.4** | **37.3 ± 0.3** | **10.2 ± 0.3** | **28.2 ± 0.9** | **39.0 ± 0.3** |
| ViT | Image prototype | 5.4 ± 0.1 | 17.5 ± 0.2 | 26.5 ± 0.5 | 5.4 ± 0.3 | **17.8 ± 0.5** | 25.7 ± 1.1 |
| | Both prototypes | **6.8 ± 0.3** | **19.2 ± 0.3** | **28.5 ± 0.4** | **6.6 ± 0.5** | 17.5 ± 0.5 | **26.9 ± 0.5** |

## C.5 PERFORMANCE COMPARISON BETWEEN BERT AND CLIP TEXT ENCODER

To ensure a fair comparison of PDS with optimization-based multimodal dataset distillation baselines, we replace the BERT encoder used in prior work with the CLIP text encoder. As shown in Table 12, this replacement not only ensures a fair comparison but also consistently improves performance.

Table 12: **Comparison of BERT and CLIP as text encoders in the 100-pair setting on Flickr30K.** Within each method, we compare BERT and CLIP as text encoders under the same vision backbone (ResNet or ViT). In all cases, replacing BERT with CLIP consistently improves performance.

| Method | Evaluation Model | IR@1 | IR@5 | IR@10 | TR@1 | TR@5 | TR@10 |
|---|---|---|---|---|---|---|---|
| TESLA-VL | ResNet+BERT | 0.3 ± 0.1 | 1.9 ± 0.2 | 3.4 ± 0.2 | 2.0 ± 0.4 | 7.9 ± 0.5 | 12.2 ± 0.5 |
| | ResNet+CLIP | **4.1 ± 0.3** | **14.7 ± 0.9** | **22.9 ± 1.2** | **6.5 ± 0.4** | **17.8 ± 1.4** | **27.3 ± 1.4** |
| | ViT+BERT | 0.2 ± 0.1 | 1.0 ± 0.2 | 1.9 ± 0.5 | 0.5 ± 0.2 | 2.0 ± 0.6 | 3.9 ± 1.4 |
| | ViT+CLIP | **2.1 ± 0.3** | **7.8 ± 0.7** | **13.1 ± 1.2** | **2.6 ± 0.6** | **8.7 ± 0.9** | **13.7 ± 1.4** |
| LoRS | ResNet+BERT | 2.0 ± 0.1 | 7.3 ± 0.3 | 11.8 ± 0.2 | 3.7 ± 0.4 | 13.4 ± 0.5 | 19.8 ± 0.8 |
| | ResNet+CLIP | **6.3 ± 0.1** | **18.6 ± 0.1** | **28.0 ± 0.2** | **9.1 ± 0.2** | **24.3 ± 0.4** | **34.5 ± 0.8** |
| | ViT+BERT | 0.4 ± 0.1 | 2.7 ± 0.1 | 5.1 ± 0.2 | 1.7 ± 0.2 | 6.2 ± 0.5 | 10.3 ± 0.5 |
| | ViT+CLIP | **2.8 ± 0.1** | **9.9 ± 0.4** | **16.1 ± 0.2** | **5.2 ± 0.3** | **13.6 ± 0.5** | **20.5 ± 0.2** |

## C.6 SENSITIVITY ANALYSIS OF THE HYPERPARAMETERS

We conduct extensive ablation studies on decoder hyperparameters such as the guidance scale and the number of sampling steps, as well as on the PDS-specific hyperparameters, including the clustering seeds and the similarity-pruning threshold ratio. We first examine the sensitivity to the decoder hyperparameters. Varying the guidance scale $\in \{3, 5, 7, 10, 15\}$ and the number of sampling steps $\in \{25, 50, 100, 150, 200\}$ yields consistent retrieval accuracy, with only slight degradation at smaller values where the effect of prototype conditioning weakens. These trends are shown in Figure 6, which reports the aggregated retrieval accuracy, computed as the average of IR@1, IR@5, IR@10, TR@1, TR@5, and TR@10, using distilled sets of 100 and 300 pairs.

We next analyze the sensitivity to the PDS-specific hyperparameters. Performance remains consistent across different clustering seeds, as illustrated by the tight box plot shown in Figure 6, using 20 different seeds. The same figure also presents the effect of the similarity-pruning threshold ratio $\in \{0.0, 0.1, 0.2, 0.3, 0.5\}$, showing that moderate pruning reliably improves performance by removing weakly aligned pairs.

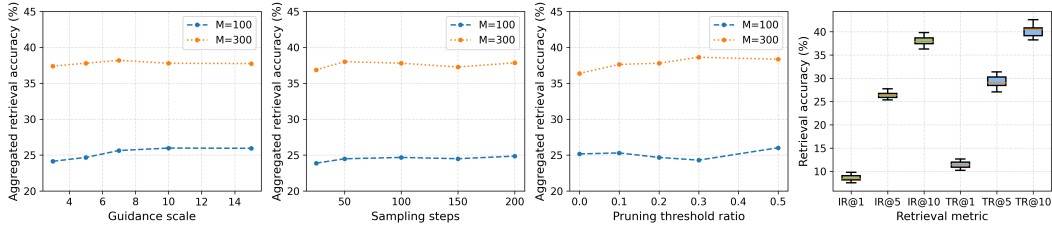

Figure 6: **Ablation study on hyperparameters.** The first three panels show the results obtained by varying the guidance scale, the number of sampling steps, and the pruning threshold ratio, respectively. Each panel reports the aggregated retrieval accuracy, computed as the average of IR@1, IR@5, IR@10, TR@1, TR@5, and TR@10, using distilled sets of 100 and 300 pairs on Flickr30K. The final panel presents a boxplot of retrieval accuracy across 20 different clustering seeds. All results in the figure were evaluated using a ResNet backbone.

## C.7 EFFECT OF THE DISTILLED DATASET SIZE ON PERFORMANCE

Table 13 presents a monotonic improvement in performance as the distilled dataset size increases, with the results gradually approaching those obtained from the full dataset. In practice, the distilled dataset size may be chosen heuristically based on the structure of the data embeddings. When the embeddings form well-separated and compact clusters, a relatively small number of pairs is often sufficient because each cluster generally corresponds to a coherent semantic group. In contrast, when the embedding space is more diffuse or spread out, using a larger number of pairs is usually beneficial, as it allows the distilled dataset to capture more local structure.

Table 13: **Increasing the distilled dataset size consistently improves performance.** We distill Flickr30K using a wide range of distilled dataset sizes and evaluate the resulting distilled datasets. In all cases, larger distilled sets lead to better performance.

| Evaluation Model | Pairs | IR@1 | IR@5 | IR@10 | TR@1 | TR@5 | TR@10 |
|---|---|---|---|---|---|---|---|
| ResNet | 100 | 7.9 ± 0.3 | 25.8 ± 0.4 | 37.3 ± 0.3 | 10.2 ± 0.3 | 28.2 ± 0.9 | 39.0 ± 0.3 |
| | 300 | 14.4 ± 0.4 | 38.1 ± 0.2 | 51.4 ± 0.4 | 18.7 ± 0.5 | 45.0 ± 0.4 | 57.8 ± 0.6 |
| | 500 | 17.1 ± 0.2 | 43.0 ± 0.2 | 56.4 ± 0.1 | 23.9 ± 0.9 | 50.5 ± 0.6 | 62.0 ± 0.3 |
| | 1000 | 19.5 ± 0.3 | 46.4 ± 0.4 | 58.9 ± 0.2 | 27.4 ± 0.4 | 55.3 ± 0.6 | 67.5 ± 0.6 |
| | 1500 | 20.5 ± 0.2 | 48.7 ± 0.3 | 62.5 ± 0.2 | 28.8 ± 0.5 | 59.2 ± 0.8 | 71.2 ± 0.4 |
| | Full Dataset | 28.5 ± 0.2 | 59.6 ± 0.1 | 71.4 ± 0.1 | 46.0 ± 0.6 | 76.2 ± 0.3 | 84.4 ± 0.2 |

## C.8 ALTERNATIVE CLIP VARIANT

There are two publicly available pre-trained unCLIP models based on CLIP ViT-L/14 and ViT-H/14, respectively. For our main experiments, we adopt CLIP ViT-L/14 because it is more computationally efficient for extracting image and text features. We additionally evaluate PDS using CLIP ViT-H/14 to examine whether the improvement generalizes to a different CLIP variant. As shown in Table 14, using CLIP ViT-H/14 results in improved performance, likely due to its stronger alignment between image and text features. Importantly, our PDS framework continues to outperform all other baselines even with this CLIP variant, indicating that the observed gains are not specific to CLIP ViT-L/14.

Table 14: **Results with an alternative CLIP variant in the 300-pair setting on Flickr30K.** We evaluate PDS using CLIP ViT-H/14 to examine whether the improvements generalize across a different variant. CLIP ViT-H/14 achieves higher performance, likely due to its stronger image-text alignment.

| Evaluation Model | CLIP variant | IR@1 | IR@5 | IR@10 | TR@1 | TR@5 | TR@10 |
|---|---|---|---|---|---|---|---|
| ResNet | ViT-L/14 | 14.4 ± 0.4 | 38.1 ± 0.2 | 51.4 ± 0.4 | 18.7 ± 0.5 | 45.0 ± 0.4 | 57.8 ± 0.6 |
| | ViT-H/14 | **17.2 ± 0.5** | **42.6 ± 0.3** | **55.6 ± 0.4** | **23.7 ± 0.9** | **47.3 ± 0.8** | **60.0 ± 0.8** |
| ViT | ViT-L/14 | 9.1 ± 0.1 | 27.3 ± 0.4 | 38.4 ± 0.4 | 9.6 ± 0.3 | 26.1 ± 0.5 | 37.5 ± 1.2 |
| | ViT-H/14 | **12.0 ± 0.1** | **32.8 ± 0.3** | **43.2 ± 0.6** | **12.7 ± 0.6** | **32.4 ± 0.2** | **44.3 ± 0.2** |

## C.9 ROBUSTNESS OF THE PDS TO RARE SAMPLES

The prototypes from PDS may be more influenced by features from dominant classes, which could lead to underrepresentation of rare or long-tail classes. However, we note that this limitation is shared by subset selection and other distillation baselines, and PDS empirically demonstrates stronger robustness in such cases. For this evaluation, we define rare samples as the 200 Flickr30K test samples farthest from the mean embedding of the training dataset. Using the 100 samples selected or distilled from Flickr30K by each method, we measure the retrieval accuracy on this rare subset. As shown in Table 15, PDS consistently outperforms both subset selection and other distillation baselines. These results indicate that, despite the inherent limitation, PDS provides a better coverage of rare examples compared to all other baselines.

Table 15: **PDS is more robust to rare samples than existing methods.** We evaluate each method on 200 rare test samples from Flickr30K, defined as those farthest from the mean embedding of the training dataset. In the 100-pair setting on Flickr30K, PDS consistently outperforms both subset selection and other distillation baselines, demonstrating stronger robustness to rare samples.

| Evaluation Model | Methods | IR@1 | IR@5 | IR@10 | TR@1 | TR@5 | TR@10 |
|---|---|---|---|---|---|---|---|
| ResNet | K-center | 8.5 ± 1.4 | 26.7 ± 0.9 | 40.0 ± 0.8 | 15.4 ± 0.9 | 40.3 ± 0.5 | 54.0 ± 1.6 |
| | Herding | 9.4 ± 0.5 | 28.7 ± 0.6 | 43.7 ± 1.7 | 15.6 ± 1.2 | 38.8 ± 2.0 | 51.3 ± 1.4 |
| | CLIP score | 7.2 ± 0.4 | 22.3 ± 0.4 | 33.5 ± 0.5 | 12.8 ± 1.6 | 30.6 ± 1.2 | 42.9 ± 2.1 |
| | LAION filtering | 6.6 ± 0.3 | 20.2 ± 0.5 | 31.8 ± 0.3 | 8.8 ± 0.5 | 24.3 ± 1.0 | 36.3 ± 1.8 |
| | Image-based | 6.7 ± 0.5 | 22.4 ± 0.3 | 33.3 ± 0.3 | 12.3 ± 1.8 | 30.2 ± 1.4 | 42.1 ± 1.5 |
| | TESLA-VL | 11.0 ± 1.7 | 35.0 ± 1.5 | 50.9 ± 2.1 | 15.0 ± 2.9 | 39.6 ± 3.9 | 52.5 ± 3.0 |
| | LoRS | 14.1 ± 0.5 | 37.2 ± 1.0 | 54.0 ± 0.5 | 22.6 ± 1.9 | 47.4 ± 1.4 | 62.2 ± 2.3 |
| | PDS | **23.2 ± 0.8** | **58.4 ± 0.6** | **74.5 ± 0.8** | **25.2 ± 1.0** | **58.8 ± 1.2** | **73.6 ± 2.7** |

## C.10 Result on the Flickr8K Dataset

We further evaluate PDS on the Flickr8K (Hodosh et al., 2013) dataset and compare it against subset selection methods. As shown in Table 16, PDS consistently outperforms the subset selection baselines.

Table 16: **Comparison of PDS with subset selection methods on Flickr8K.** We evaluate PDS and subset selection baselines in the 100-pair setting on Flickr8K, and PDS consistently outperforms all subset selection methods.

| Evaluation Model | Methods | IR@1 | IR@5 | IR@10 | TR@1 | TR@5 | TR@10 |
|---|---|---|---|---|---|---|---|
| ResNet | K-center | 0.8 ± 0.8 | 3.4 ± 3.1 | 6.1 ± 5.2 | 1.2 ± 1.3 | 3.9 ± 3.5 | 6.6 ± 5.1 |
| | Herding | 0.8 ± 0.9 | 3.3 ± 3.4 | 5.6 ± 5.5 | 1.0 ± 1.4 | 3.8 ± 4.4 | 6.0 ± 6.5 |
| | CLIP score | 3.5 ± 0.2 | 12.5 ± 0.2 | 19.4 ± 0.3 | 5.5 ± 0.4 | 14.4 ± 0.6 | 22.1 ± 0.5 |
| | LAION filtering | 3.5 ± 0.1 | 13.0 ± 0.3 | 20.4 ± 0.3 | 6.4 ± 0.5 | 15.2 ± 0.9 | 22.6 ± 0.7 |
| | Image-based | 3.4 ± 0.2 | 12.3 ± 0.2 | 18.8 ± 0.2 | 5.6 ± 0.3 | 14.1 ± 0.6 | 21.9 ± 0.4 |
| | PDS | **8.5 ± 0.2** | **27.0 ± 0.3** | **39.8 ± 0.3** | **10.3 ± 0.2** | **27.1 ± 0.7** | **38.3 ± 0.4** |

## C.11 Comparison of Text, Image, and Average Prototype Conditioning in the UnCLIP Decoder

We additionally generate images conditioned either on the text prototype or on the average of the text and image prototypes. As shown in Table 17, conditioning on the text prototype results in degraded performance, while averaging the text and image prototypes produces results comparable to conditioning on the image prototype, with no noticeable improvements.

Table 17: **Comparison of conditioning on text, image, and average prototypes.** We evaluate different conditioning strategies in the 100-pair setting on Flickr30K. Conditioning on the text prototype degrades performance, while using the average of the text and image prototypes achieves results comparable to conditioning on the image prototype, providing no additional gains.

| Evaluation Model | Methods | IR@1 | IR@5 | IR@10 | TR@1 | TR@5 | TR@10 |
|---|---|---|---|---|---|---|---|
| ResNet | Text prototype | **8.1 ± 0.2** | 24.9 ± 0.2 | 36.6 ± 0.4 | 10.8 ± 0.4 | **28.2 ± 0.7** | 38.3 ± 0.7 |
| | Average prototype | 7.9 ± 0.2 | **26.2 ± 0.5** | **37.9 ± 0.5** | **11.1 ± 0.3** | **28.2 ± 0.5** | **39.4 ± 0.3** |
| | Image prototype | 7.9 ± 0.3 | 25.8 ± 0.4 | 37.3 ± 0.3 | 10.2 ± 0.3 | **28.2 ± 0.9** | 39.0 ± 0.3 |
| ViT | Text prototype | 5.6 ± 0.2 | 17.3 ± 0.2 | 26.4 ± 0.1 | 4.6 ± 0.4 | 16.3 ± 0.7 | 23.9 ± 0.6 |
| | Average prototype | 5.9 ± 0.1 | 18.5 ± 0.1 | 27.7 ± 0.3 | 6.5 ± 0.6 | **18.4 ± 0.3** | **27.3 ± 0.3** |
| | Image prototype | **6.8 ± 0.3** | **19.2 ± 0.3** | **28.5 ± 0.4** | **6.6 ± 0.5** | 17.5 ± 0.5 | 26.9 ± 0.5 |

## C.12 Comparison of Joint and Separate Clustering Approaches

It is certainly possible to perform clustering directly in a joint embedding space, but the two modalities can differ in how their embeddings are distributed, which can sometimes cause the clustering to be influenced more by one modality than the other. For this reason, we cluster each modality separately and then align the resulting clusters across modalities. As shown in Table 18, the two approaches perform comparably, and the separate clustering approach is slightly better in a few cases.

Table 18: **Comparison of joint-embedding clustering (Joint) and separate clustering (Separate).** We evaluate different clustering strategies by distilling the Flickr30K and MS-COCO into 100 and 300 pairs, and evaluating the resulting distilled sets using a ResNet backbone. Overall, both clustering strategies perform comparably, with the separate clustering approach showing slight improvements in a few cases.

| Dataset | Pairs | Methods | IR@1 | IR@5 | IR@10 | TR@1 | TR@5 | TR@10 |
|---|---|---|---|---|---|---|---|---|
| Flickr30K | 100 | Joint | **8.3** ± 0.1 | 25.3 ± 0.2 | 36.8 ± 0.4 | **11.0** ± 0.4 | **29.5** ± 0.4 | **40.7** ± 0.4 |
| | | Separate | 7.9 ± 0.3 | **25.8** ± 0.4 | **37.3** ± 0.3 | 10.2 ± 0.3 | 28.2 ± 0.9 | 39.0 ± 0.3 |
| | 300 | Joint | **14.6** ± 0.5 | **38.8** ± 0.3 | **52.3** ± 0.4 | 17.5 ± 0.4 | 43.5 ± 0.4 | 56.4 ± 0.8 |
| | | Separate | 14.4 ± 0.4 | 38.1 ± 0.2 | 51.4 ± 0.4 | **18.7** ± 0.5 | **45.0** ± 0.4 | **57.8** ± 0.6 |
| MS-COCO | 100 | Joint | 2.5 ± 0.1 | **10.2** ± 0.2 | **17.5** ± 0.3 | 4.0 ± 0.1 | 12.9 ± 0.2 | 20.6 ± 0.2 |
| | | Separate | **2.8** ± 0.1 | 10.0 ± 0.2 | 17.3 ± 0.3 | **4.5** ± 0.2 | **14.0** ± 0.3 | **21.4** ± 0.4 |
| | 300 | Joint | 4.9 ± 0.1 | 16.6 ± 0.2 | 25.9 ± 0.3 | 6.6 ± 0.2 | 19.3 ± 0.3 | 29.2 ± 0.4 |
| | | Separate | **5.3** ± 0.2 | **17.2** ± 0.4 | **27.2** ± 0.6 | **7.4** ± 0.3 | **20.7** ± 0.3 | **30.2** ± 0.4 |

# D APPLICATION

## D.1 APPLICATION OF THE DISTILLED DATASET TO ASIF

We apply our distilled dataset to ASIF (Norelli et al., 2023). Although ASIF and PDS are developed for different purposes, they are complementary. ASIF aligns pre-trained unimodal encoders without additional training by computing relative representations whose elements are cosine similarities between a sample and each anchor in the anchor set. Consequently, the quality of the anchor set is crucial, and the computational cost of ASIF increases with the number of anchors. The original ASIF paper reports that more than one million anchors are required to reach CLIP-level performance, highlighting the importance of identifying compact yet informative anchors.

PDS can address this point by producing a small set of highly informative samples that can serve as stronger anchors than those obtained by selecting subsets of the original dataset. In Table 2 of the manuscript, we already show that PDS produces more representative samples than subset selection baselines. Based on this result, we evaluate the performance of ASIF with 300 anchors obtained from Flickr30K using (i) subset selection methods, (ii) other distillation methods, and (iii) PDS. For this comparison, we use a ResNet image encoder and a CLIP text encoder to compute the relative representations, and performance is evaluated through retrieval accuracy on Flickr30K. As shown in Table 19, the anchors produced by PDS lead to the strongest performance among all compared methods, implying that PDS allows ASIF to operate effectively with fewer anchors.

Table 19: **Application of the distilled dataset as anchors in ASIF.** To evaluate how well distilled or selected datasets can serve as compact anchors for ASIF, we construct 300 anchors from Flickr30K for each method, including subset selection, other distillation approaches, and PDS. Using a ResNet image encoder and a CLIP text encoder to compute relative representations, we measure retrieval accuracy on Flickr30K. Anchors produced by PDS consistently outperform those obtained from all other baselines, demonstrating that PDS yields more informative and effective anchors for ASIF.

| Evaluation Model | Methods | IR@1 | IR@5 | IR@10 | TR@1 | TR@5 | TR@10 |
|---|---|---|---|---|---|---|---|
| ResNet | K-center | 3.8 | 14.4 | 24.2 | 4.0 | 14.8 | 22.7 |
| | Herding | 6.0 | 21.5 | 32.5 | 6.9 | 19.1 | 28.1 |
| | CLIP score | 2.1 | 9.5 | 16.9 | 3.5 | 11.5 | 18.7 |
| | LAION filtering | 3.1 | 11.2 | 17.9 | 3.4 | 12.5 | 17.4 |
| | Image-based | 2.3 | 9.6 | 17.1 | 4.1 | 12.6 | 18.3 |
| | TESLA-VL | 5.3 | 19.4 | 29.8 | 5.0 | 16.8 | 25.9 |
| | LoRS | 4.9 | 18.6 | 28.2 | 5.7 | 16.6 | 25.9 |
| | PDS | **9.9** | **29.8** | **41.8** | **9.6** | **27.7** | **38.6** |

## D.2 INTEGRATION OF PDS INTO OPTIMIZATION-BASED MULTIMODAL DISTILLATION

We integrate PDS into existing optimization-based multimodal distillation methods by using the PDS-distilled dataset as an initialization, which is known to be crucial for optimization-based distillation. Although replacing real samples used for initialization with the PDS-distilled dataset provides a more informative starting point, the subsequent optimization process leads to an architecture-dependent distilled dataset, which undermines the cross-architecture generalization achieved by PDS. As shown in Table 20, initializing the optimization-based method with the PDS-distilled dataset tends to improve its performance, but it still underperforms compared to using PDS alone.

Table 20: **Integrating PDS into optimization-based distillation baselines provides little advantage.** We evaluate optimization-based multimodal distillation methods initialized with the PDS-distilled dataset in the 100-pair setting on Flickr30K. Although this initialization provides a more informative starting point, the subsequent optimization makes the distilled dataset architecture-dependent, limiting the cross-architecture generalization achieved by PDS. Consequently, while PDS initialization tends to improve these baselines, this integrated approach still underperforms compared to using PDS alone.

| Evaluation Model | Method | IR@1 | IR@5 | IR@10 | TR@1 | TR@5 | TR@10 |
|---|---|---|---|---|---|---|---|
| ResNet | TESLA-VL | 4.1 ± 0.3 | 14.7 ± 0.9 | 22.9 ± 1.2 | 6.5 ± 0.4 | 17.8 ± 1.4 | 27.3 ± 1.4 |
| | TESLA-VL +PDS | 5.2 ± 0.4 | 17.7 ± 0.6 | 27.5 ± 0.4 | 7.5 ± 1.0 | 22.4 ± 1.3 | 33.1 ± 1.3 |
| | LoRS | 6.3 ± 0.1 | 18.6 ± 0.1 | 28.0 ± 0.2 | 9.1 ± 0.2 | 24.3 ± 0.4 | 34.5 ± 0.8 |
| | LoRS+PDS | 5.7 ± 0.2 | 19.5 ± 0.6 | 29.2 ± 0.9 | 8.3 ± 1.2 | 24.1 ± 1.3 | 34.3 ± 1.4 |
| | PDS | **7.9 ± 0.3** | **25.8 ± 0.4** | **37.3 ± 0.3** | **10.2 ± 0.3** | **28.2 ± 0.9** | **39.0 ± 0.3** |

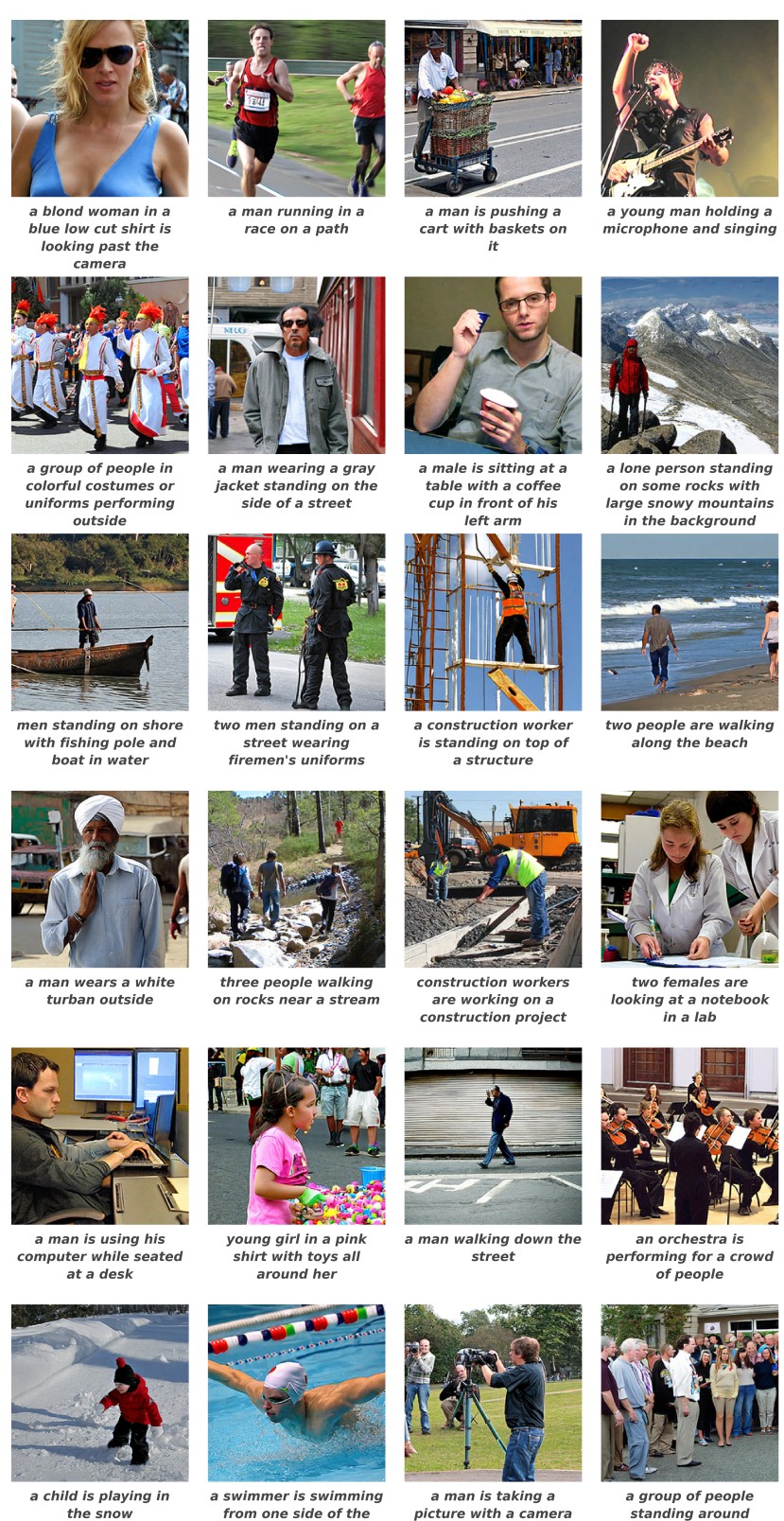

Figure 7: Examples of synthetic images and retrieved captions in the 100-pair setting on Flickr30K.

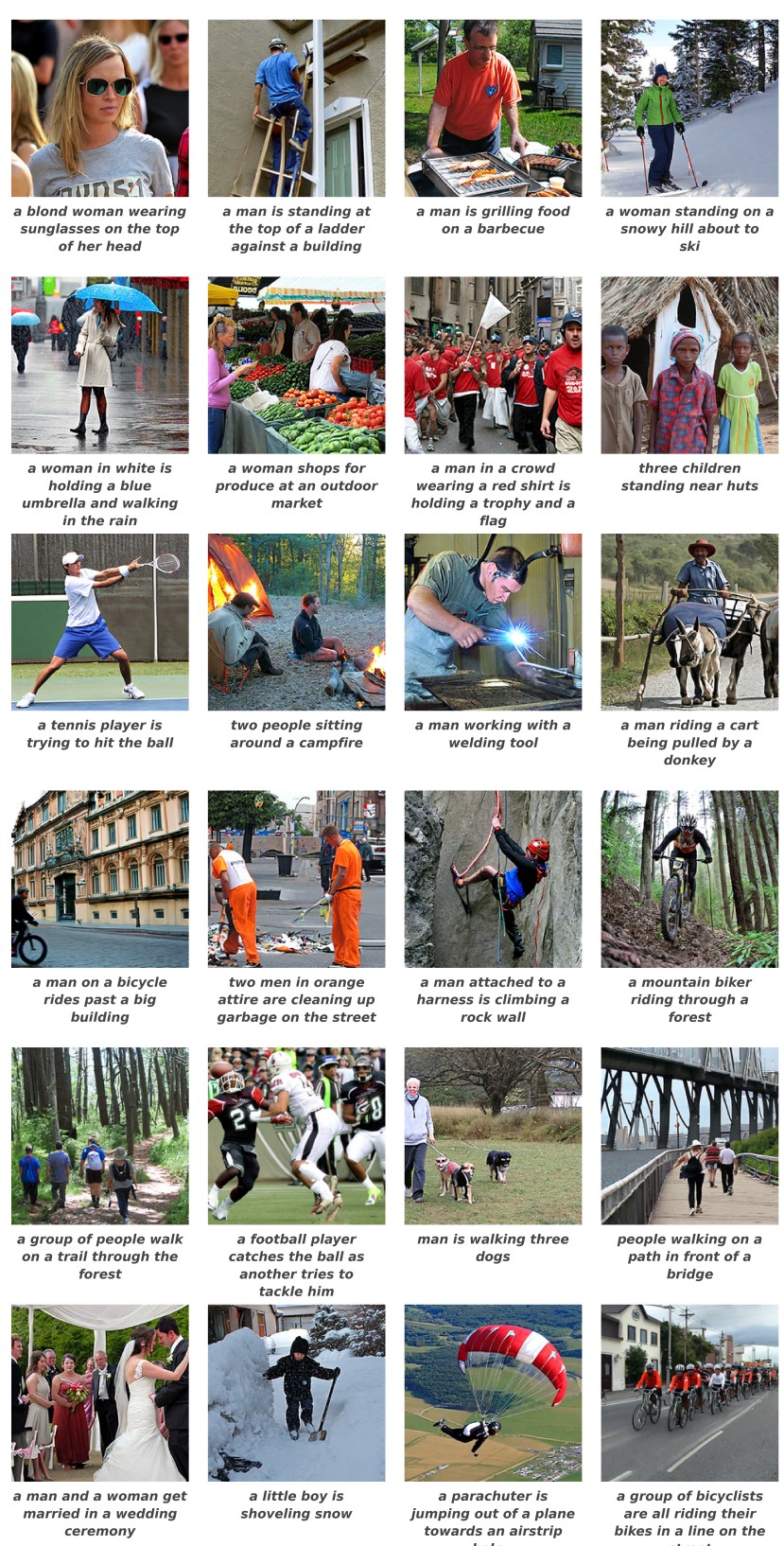

Figure 8: Examples of synthetic images and retrieved captions in the 300-pair setting on Flickr30K.

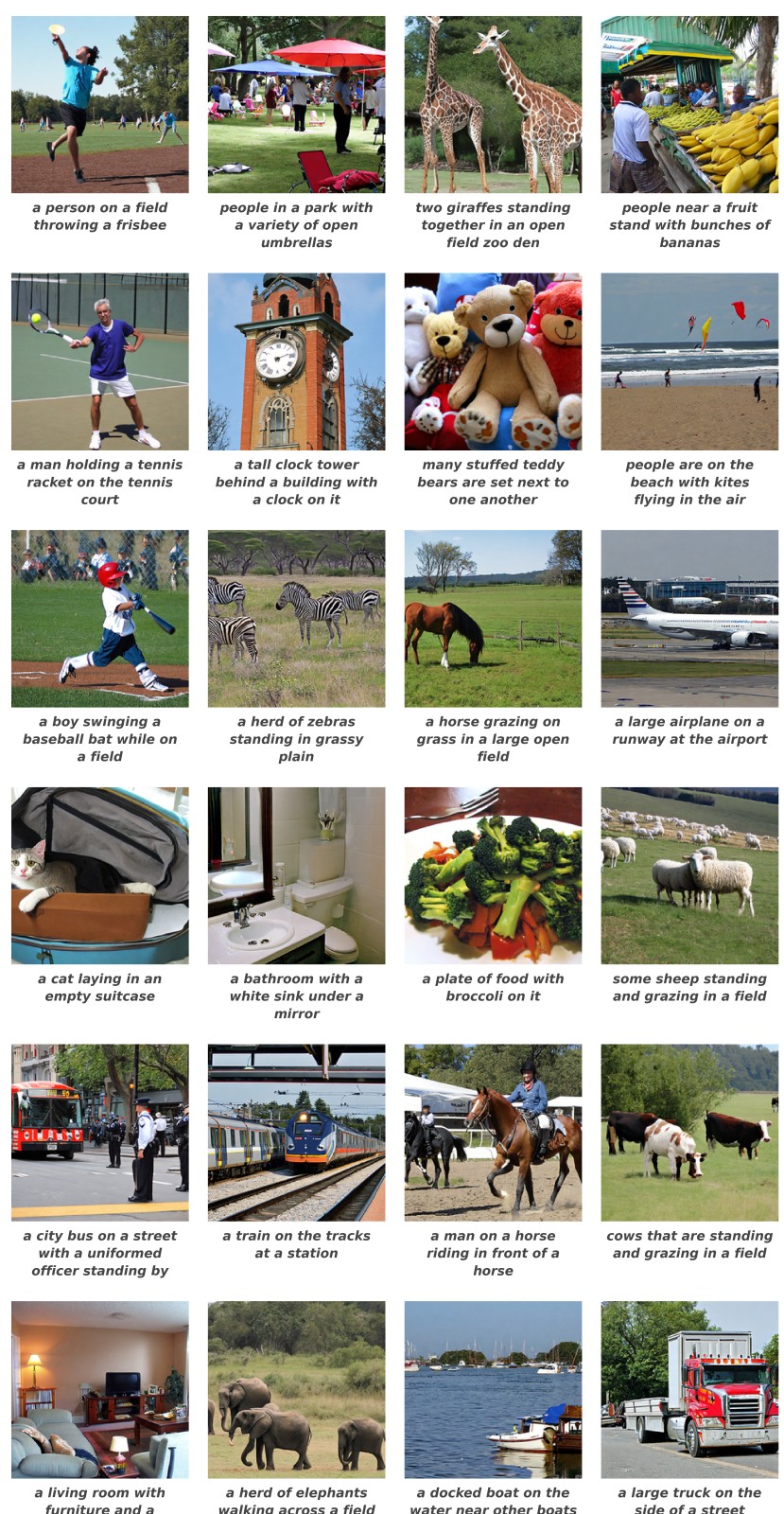

Figure 9: Examples of synthetic images and retrieved captions in the 100-pair setting on MS-COCO.

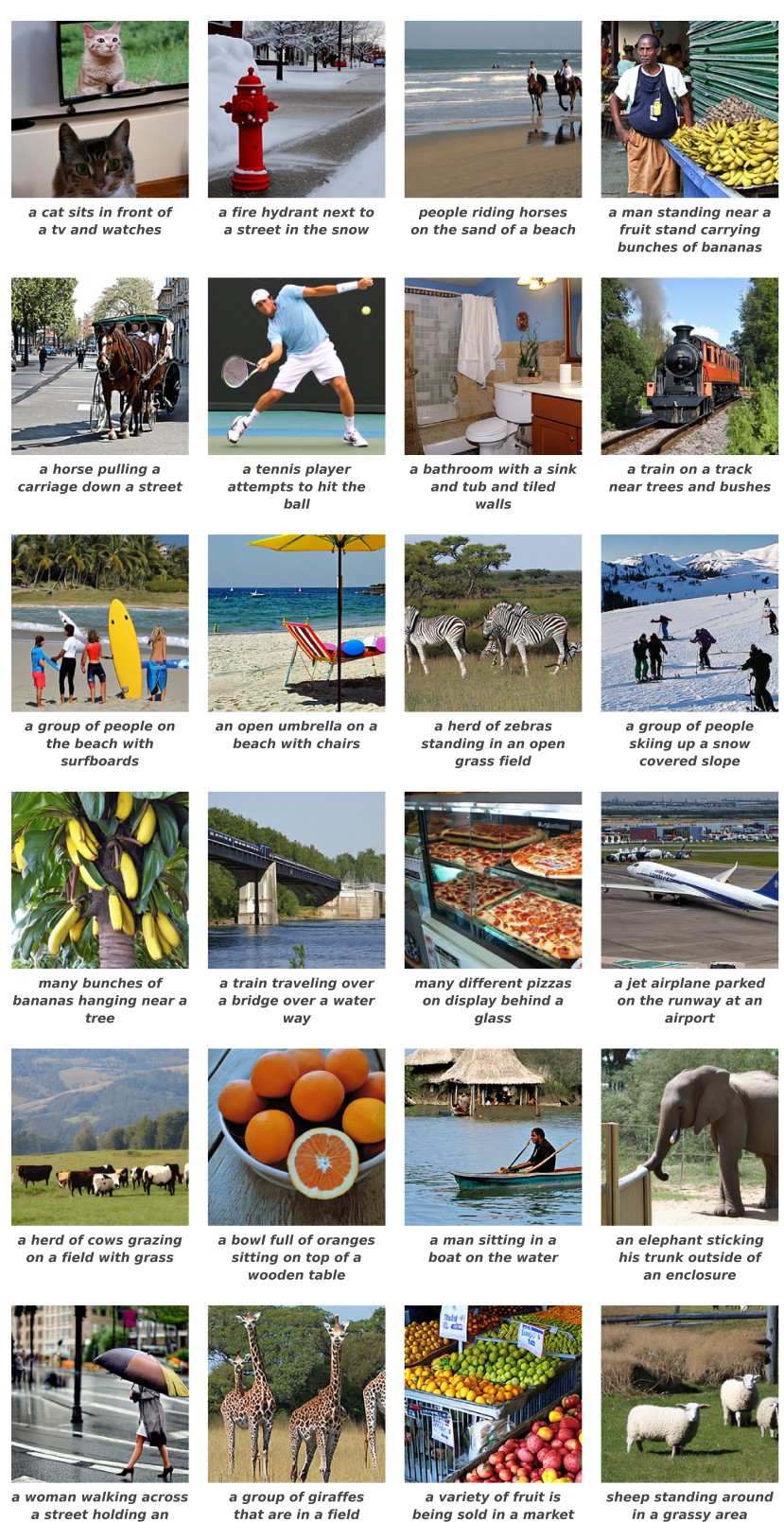

Figure 10: Examples of synthetic images and retrieved captions in the 300-pair setting on MS-COCO.

