# OpenReview forum: "Multimodal Dataset Distillation Made Simple by Prototype-Guided Data Synthesis"
_ICLR.cc/2026/Conference — ICLR 2026 Poster_

### Official Review · Reviewer_gG1N · 2025-10-31

**Soundness:** 2
**Presentation:** 3
**Contribution:** 2
**Rating:** 4
**Confidence:** 2

**Summary:**

This paper proposes a prototype-guided data synthesis framework for multimodal dataset distillation that removes the need for large-scale training and optimization. It achieves efficient and scalable multimodal data generation, outperforming previous optimization-based methods while improving cross-architecture generalization.

**Strengths:**

1. The proposed method is learning-free, making it simple, efficient, and architecture-independent.
2. The framework works well across different architectures, showing cross-architecture generalization.
3. It synthesizes semantically aligned image–text pairs, improving multimodal learning performance.

**Weaknesses:**

1. Will averaging embeddings to construct prototypes cause degraded representations?

2. The method uses the unCLIP decoder to reconstruct images from CLIP embeddings/text. The capibility of the decoder could be essential. Do authors observe failure cases and how to process them?

3. Besides Flickr30K and MS-COCO, have authors considered other data?

4. While the major classes are well represented in the proposed prototypes, have authors considered the performance on rare or long-tail classes?

**Questions:**

Please refer to 'Weaknesses'

1. Will averaging embeddings to construct prototypes cause degraded representations?

2. The method uses the unCLIP decoder to reconstruct images from CLIP embeddings/text. The capibility of the decoder could be essential. Do authors observe failure cases and how to process them?

3. Besides Flickr30K and MS-COCO, have authors considered other data?

4. While the major classes are well represented in the proposed prototypes, have authors considered the performance on rare or long-tail classes?

---

> ### Author Response · Authors · 2025-11-20
> **Response to Reviewer gG1N**
>
> Dear reviewer gG1N,
>
> Thank you for your valuable feedback and comments. We appreciate your recognition of the simplicity and efficiency of our method, its strong cross-architecture generalization, and its synthesis of semantically aligned image–text pairs. We will address your questions and concerns in the response below.
>
> **[W1 and Q1] Representation quality under embedding averaging**
>
> Averaging embeddings to construct prototypes does not degrade the representation. Instead, averaging embeddings within the same cluster produces prototypes that interpolate across the diverse semantics. As shown in Figure 3 of the manuscript, images generated from these prototypes are semantically enriched. This semantic interpolation also contributes to the performance advantage of PDS over methods that select subsets of real data, as reported in Table 2 of the manuscript.
>
> **[W2 and Q2] Failure cases of the unCLIP decoder**
>
> Failures can arise when distilling data from domains that differ significantly from those used to train CLIP and the unCLIP decoder. Since both models are primarily trained on natural images, they are less effective in specialized domains such as medical imagery, and image generation from the corresponding embeddings may fail. For such domains, fine-tuning both the CLIP and the unCLIP decoder becomes necessary. We have added this point to the limitations section on page 10 of the revised manuscript.
>
> **[W3 and Q3] Evaluation on additional dataset**
>
> We further evaluated PDS on the Flickr8K [1] dataset and compared it against subset selection methods. We distilled Flickr8K into 100 samples and evaluated the distilled data using a ResNet-50 backbone.
> As shown in the table below, PDS consistently outperforms the subset selection baselines. We have added these results to Appendix C.10 of the revised manuscript.
> \begin{array}{lcccccc}
> \hline
> \textbf{Methods}& \textbf{IR@1} & \textbf{IR@5} & \textbf{IR@10}& \textbf{TR@1} & \textbf{TR@5} & \textbf{TR@10} \newline
> \hline
> \text{K-center}& 0.8 \pm 0.8 & 3.4 \pm 3.1 & 6.1 \pm 5.2& 1.2 \pm 1.3 & 3.9 \pm 3.5 & 6.6 \pm 5.1 \newline
> \text{Herding}& 0.8 \pm 0.9 & 3.3 \pm 3.4 & 5.6 \pm 5.5& 1.0 \pm 1.4 & 3.8 \pm 4.4 & 6.0 \pm 6.5 \newline
> \text{CLIP score}& 3.5 \pm 0.2 & 12.5 \pm 0.2 & 19.4 \pm 0.3& 5.5 \pm 0.4 & 14.4 \pm 0.6 & 22.1 \pm 0.5 \newline
> \text{LAION filtering}& 3.5 \pm 0.1 & 13.0 \pm 0.3 & 20.4 \pm 0.3& 6.4 \pm 0.5 & 15.2 \pm 0.9 & 22.6 \pm 0.7 \newline
> \text{Image-based}
> & 3.4 \pm 0.2 & 12.3 \pm 0.2 & 18.8 \pm 0.2& 5.6 \pm 0.3 & 14.1 \pm 0.6 & 21.9 \pm 0.4 \newline
> \textbf{PDS}& \mathbf{8.5 \pm 0.2} & \mathbf{27.0 \pm 0.3} & \mathbf{39.8 \pm 0.3}& \mathbf{10.3 \pm 0.2} & \mathbf{27.1 \pm 0.7} & \mathbf{38.3 \pm 0.4} \newline
> \hline
> \end{array}
> [1] Hodosh et al. Framing Image Description as a Ranking Task. Journal of Artificial Intelligence Research 2013.

---

> ### Author Response · Authors · 2025-11-20
> **Response to Reviewer gG1N**
>
> **[W4 and Q4] Performance on rare and long-tail samples**
>
> We acknowledge that the proposed prototypes may be more influenced by features from major classes, which could lead to underrepresentation of rare or long-tail classes. We have added this point to the limitations section on page 10 of the revised manuscript.
>
> However, we note that this limitation is also present in subset selection methods and other distillation baselines, and PDS empirically demonstrates stronger robustness under such conditions. For this evaluation, we defined rare samples as the 200 Flickr30K test samples farthest from the mean embedding of the training data. Using the 100 samples distilled from Flickr30K by each method, we measured retrieval accuracy on these rare samples with a ResNet-50 backbone.
> As shown in the table below, PDS consistently outperforms both subset selection methods and other distillation approaches on this rare subset. These results indicate that, despite the inherent limitation, PDS provides a better coverage of rare examples compared to existing baselines. We have added this analysis to Appendix C.9 of the revised manuscript.
> \begin{array}{lcccccc}
> \hline
> \textbf{Methods}
> & \textbf{IR@1} & \textbf{IR@5} & \textbf{IR@10}& \textbf{TR@1} & \textbf{TR@5} & \textbf{TR@10} \newline\hline
> \text{K-center} & 8.5 \pm 1.4 & 26.7 \pm 0.9 & 40.0 \pm 0.8 & 15.4 \pm 0.9 & 40.3 \pm 0.5 & 54.0 \pm 1.6 \newline
> \text{Herding} & 9.4 \pm 0.5 & 28.7 \pm 0.6 & 43.7 \pm 1.7 & 15.6 \pm 1.2 & 38.8 \pm 2.0 & 51.3 \pm 1.4 \newline
> \text{CLIP score} & 7.2 \pm 0.4 & 22.3 \pm 0.4 & 33.5 \pm 0.5 & 12.8 \pm 1.6 & 30.6 \pm 1.2 & 42.9 \pm 2.1 \newline
> \text{LAION filtering} & 6.6 \pm 0.3 & 20.2 \pm 0.5 & 31.8 \pm 0.3 & 8.8 \pm 0.5 & 24.3 \pm 1.0 & 36.3 \pm 1.8 \newline
> \text{Image-based} & 6.7 \pm 0.5 & 22.4 \pm 0.3 & 33.3 \pm 0.3 & 12.3 \pm 1.8 & 30.2 \pm 1.4 & 42.1 \pm 1.5 \newline
> \text{TESLA-VL} & 11.0 \pm 1.7 & 35.0 \pm 1.5  & 50.9 \pm 2.1 & 15.0 \pm 2.9 & 39.6 \pm 3.9  & 52.5 \pm 3.0  \newline
> \text{LoRS} & 14.1 \pm 0.5 & 37.2 \pm 1.0 & 54.0 \pm 0.5 & 22.6 \pm 1.9 & 47.4 \pm 1.4 & 62.2 \pm 2.3 \newline
> \textbf{PDS} & \mathbf{23.2 \pm 0.8} & \mathbf{58.4 \pm 0.6} & \mathbf{74.5 \pm 0.8} & \mathbf{25.2 \pm 1.0} & \mathbf{58.8 \pm 1.2} & \mathbf{73.6 \pm 2.7} \newline
> \hline
> \end{array}
> Please let us know if you have any further concerns.
>
> Sincerely,
> Authors

---

### Official Review · Reviewer_x6Kp · 2025-11-03

**Soundness:** 3
**Presentation:** 3
**Contribution:** 3
**Rating:** 6
**Confidence:** 4

**Summary:**

The paper proposes a learning-free framework for multimodal dataset distillation, which avoids complex optimization by leveraging pre-trained models such as CLIP and unCLIP generation models, with new designs. The experiments validate the effectiveness.

**Strengths:**

1. The intuition is clear and novel. Instead of using costly optimization-based approaches for dataset distillation, the paper propose to reuse pre-trained models such as CLIP, unCLIP to derive the underlying multimodal embeddings, and use that for later dataset generation. This provides an inspiring direction  for multimodal dataset distillation.
2. The results show that PDS outperforms other baselines that are optimization-based, which shows good cross-architecture generalisation ability.

**Weaknesses:**

1. The "learning-free" claim needs qualification. While the distillation process itself avoids optimization, the framework is critically dependent on specific pre-trained models: CLIP and an unCLIP decoder. This reliance on a specialized generative model capable of conditioning on CLIP image embeddings may hinder the method's general usability.
2. The model may loss ability to use newer embeddings since it requires an generative model that can accept this embedding as as condition. This constraint may limit the model's performance.

**Questions:**

1. The results show a large effect of the distilled size ($M$) on performance (comparing $M=100$ and $M=300$). Can the authors provide a broader analysis of performance scaling as $M$ increases? More importantly, what practical guidance should be used to select a suitable $M$ for a new distillation problem?

2. The method relies heavily on pre-trained models (CLIP and unCLIP). It is unclear how much of the success is due to the PDS framework versus the prior knowledge embedded in these large foundation models. Could the authors comment on this dependency?

3. Could the proposed Learning-Free PDS approach be integrated with existing optimization-based methods? Combining PDS's efficient initialization with the precision of optimization could lead to a new state-of-the-art in distilled dataset quality.

---

> ### Author Response · Authors · 2025-11-20
> **Response to Reviewer x6Kp**
>
> Dear reviewer x6Kp,
>
> Thank you for your valuable feedback and comments. We appreciate your recognition of the clarity and novelty of our intuition, as well as your assessment of the strong performance and cross-architecture generalization ability of PDS. We will address your questions and concerns in the response below.
>
> **[W1] Qualification of the learning-free claim**
>
> Our use of the term "learning-free" refers to the fact that no additional training or parameter updates are performed when distilling a new dataset. Although PDS uses pre-trained models such as CLIP and the unCLIP decoder, the distillation procedure itself is performed entirely without training.
>
> A similar naming convention appears in other settings. For instance, zero-shot generalization in large language models is still described as "zero-shot" even though pre-training may expose the model to data that implicitly contains information relevant to the downstream task. The term emphasizes that no task-specific training is performed. In the same way, PDS carries out no further optimization during distillation and thus can be regarded as learning-free.
>
> Although our current demonstration uses CLIP embeddings and an unCLIP decoder, the PDS framework itself is not restricted to CLIP. Any multimodal model that provides well-aligned image/text embeddings and a generative model capable of conditioning on those embeddings can be easily incorporated into PDS without modifying the framework.
>
> **[W2] Decoder limitation**
>
> We agree that our current implementation relies on a generative model that supports a specific embedding space, and we have explicitly acknowledged this limitation in the manuscript.
>
> Importantly, this dependency does not constrain the PDS framework itself. As more powerful generative models supporting newer embedding representations become available, they can be incorporated into PDS without modifying the overall pipeline, and we expect performance to improve accordingly. In this sense, the method is forward-compatible and can naturally benefit from advances in generative modeling.
>
> **[Q1] Performance scaling and practical choice of $M$**
>
> Our additional experiments show that performance improves consistently as $M$ increases. To examine this effect, we distilled Flickr30K using a wide range of $M$ and evaluated each distilled set using a ResNet-50 backbone. As shown in the table below, larger values of the $M$ consistently yield higher performance. We have added this ablation study to Appendix C.7 of the revised manuscript.
> \begin{array}{ccccccc}
> \hline
> \textbf{Pairs}
> & \textbf{IR@1} & \textbf{IR@5} & \textbf{IR@10}
> & \textbf{TR@1} & \textbf{TR@5} & \textbf{TR@10} \newline
> \hline
> 100  & 7.9 \pm 0.3 & 25.8 \pm 0.4 & 37.3 \pm 0.3
>      & 10.2 \pm 0.3 & 28.2 \pm 0.9 & 39.0 \pm 0.3 \newline
> 300  & 14.4 \pm 0.4 & 38.1 \pm 0.2 & 51.4 \pm 0.4  & 18.7 \pm 0.5 & 45.0 \pm 0.4 & 57.8 \pm 0.6 \newline
> 500  & 17.1 \pm 0.2 & 43.0 \pm 0.2 & 56.4 \pm 0.1 & 23.9 \pm 0.9 & 50.5 \pm 0.6 & 62.0 \pm 0.3 \newline
> 1000 & 19.5 \pm 0.3 & 46.4 \pm 0.4 & 58.9 \pm 0.2 & 27.4 \pm 0.4 & 55.3 \pm 0.6 & 67.5 \pm 0.6 \newline
> 1500 & 20.5 \pm 0.2 & 48.7 \pm 0.3 & 62.5 \pm 0.2 & 28.8 \pm 0.5 & 59.2 \pm 0.8 & 71.2 \pm 0.4 \newline
> \text{Full Dataset} & 28.5 \pm 0.2 & 59.6 \pm 0.1 & 71.4 \pm 0.1 & 46.0 \pm 0.6 & 76.2 \pm 0.3 & 84.4 \pm 0.2 \newline
> \hline
> \end{array}
> For selecting $M$ in practice, we recommend choosing it heuristically based on the structure of the data embeddings. When the embeddings form well-separated and compact clusters, a relatively small $M$ is often sufficient because each cluster generally corresponds to a coherent semantic group. In contrast, when the embedding space is more diffuse or spread out, using a larger $M$ is usually beneficial, as it allows the distilled dataset to capture more local structure.

---

> ### Author Response · Authors · 2025-11-20
> **Response to Reviewer x6Kp**
>
> **[Q2] Contribution of PDS beyond pre-trained models**
>
> We acknowledge that PDS benefits from the well-aligned and semantically rich embeddings provided by pretrained multimodal models such as CLIP. These representations are crucial for learning free multimodal dataset distillation, as we do not perform additional training to establish modality alignment. However, access to such embeddings alone does not resolve the distillation problem, and previous work has not explored how to effectively use these representations for this purpose.
> Indeed, as shown in Table 2 of the manuscript, directly selecting a subset of samples using CLIP embeddings performs worse than our method, confirming that the embeddings themselves are not sufficient.
>
> Our contribution is to introduce a framework that systematically uses these pretrained components. The framework specifies how to construct aligned image-text prototypes and how to generate synthetic samples that capture the semantic information encoded in these prototypes. Importantly, although the unCLIP decoder is able to synthesize images from CLIP embeddings, it cannot be used meaningfully for dataset distillation without the prototypes provided by PDS. Conditioning the decoder on the CLIP image embeddings of real samples may lead to lower performance, as these embeddings may not capture the semantic diversity required for distillation. The prototypes constructed by PDS supply this semantic information, enabling the decoder to generate semantically enriched images.
>
> Therefore, the effectiveness of PDS does not come from the pretrained models alone. Rather, PDS provides the mechanism that makes the knowledge in foundation models usable for multimodal dataset distillation, and to the best of our knowledge this direction has not been explored in prior research.
>
> **[Q3] Integration with optimization-based methods**
>
> Yes, PDS can be integrated into existing optimization-based methods by using the PDS-distilled dataset as an initialization. Initialization is known to be crucial for optimization-based distillation, and prior work has initialized distilled data with real samples rather than random noise. Replacing real samples with PDS-distilled data provides an even more informative starting point.
>
> However, the subsequent optimization process leads to architecture-dependent distilled data, which undermines the cross-architecture generalization achieved by PDS. As shown in the table below, initializing the optimization method with PDS tends to improve its performance, but it still underperforms compared to using PDS alone. For this evaluation, we distilled Flickr30K into 100 samples and assessed the distilled data using a ResNet-50 backbone. We have added this application to Appendix D.2 of the revised manuscript.
> \begin{array}{llcccccc}
> \hline
> \textbf{Method}
> & \textbf{IR@1} & \textbf{IR@5} & \textbf{IR@10}
> & \textbf{TR@1} & \textbf{TR@5} & \textbf{TR@10} \newline
> \hline
> \text{TESLA-VL}
> & 4.1 \pm 0.3 & 14.7 \pm 0.9 & 22.9 \pm 1.2
> & 6.5 \pm 0.4 & 17.8 \pm 1.4 & 27.3 \pm 1.4 \newline
> \text{TESLA-VL+PDS}
> & 5.2 \pm 0.4 & 17.7 \pm 0.6 & 27.5 \pm 0.4
> & 7.5 \pm 1.0 & 22.4 \pm 1.3 & 33.1 \pm 1.3 \newline
> \text{LoRS}
> & 6.3 \pm 0.1 & 18.6 \pm 0.1 & 28.0 \pm 0.2
> & 9.1 \pm 0.2 & 24.3 \pm 0.4 & 34.5 \pm 0.8 \newline
> \text{LoRS+PDS}
> & 5.7 \pm 0.2 & 19.5 \pm 0.6 & 29.2 \pm 0.9
> & 8.3 \pm 1.2 & 24.1 \pm 1.3 & 34.3 \pm 1.4 \newline
> \textbf{PDS}
> & \mathbf{7.9 \pm 0.3} & \mathbf{25.8 \pm 0.4} & \mathbf{37.3 \pm 0.3}
> & \mathbf{10.2 \pm 0.3} & \mathbf{28.2 \pm 0.9} & \mathbf{39.0 \pm 0.3} \newline
> \hline
> \end{array}
>
> Please let us know if you have any further concerns.
>
> Sincerely,
> Authors

---

### Official Review · Reviewer_WZ3d · 2025-11-06

**Soundness:** 3
**Presentation:** 3
**Contribution:** 2
**Rating:** 6
**Confidence:** 4

**Summary:**

The paper proposes an approach to dataset generation in order to train multi-modal (text and image) models on a more compact amount of data, thus requiring less compute and energy. Different from existing distillation methods based on the optimization paths of the models, this paper looks at the latent space of an existing multimodal model CLIP and generates paired (image-text) samples directly from it after clustering. The experimental evaluation shows advancement over existing methods of dataset distillation.

**Strengths:**

The work proposes a strong case for need of more compact and easier produced datasets for multi-model training. The provided experiments are extensive and include possible ablation studies.

**Weaknesses:**

I am wondering if it is possible to prove in any way, that the learning result from such distilled\synthesized dataset is similar to the original learning result. The resulting performance is only a weak sign of similarity of the trained models.

Also, I find it contradictory, that the motivation for the proposed method is based on the need for large-scale dataset existing beforehand, while this method as well requires trained CLIP-model, which means that this large scale dataset was already used as well.

Finally, I find the argument about generating samples that are sufficiently different from the copyright prohibited images rather weak. First of all, it does not remove the copyright prohibited images from the original CLIP model, second there were no strict metrics provided proving that it is indeed the case that images would sufficiently differ compared to other methods (except for one Figure).

Minor:

- Referncing to tables and figures in Section3 is a bit cryptic - maybe more details on how exactly table proves the point would be helpful for the reader, who does not have to go to the tables beforehand then.

**Questions:**

1 - Why sampling inputs from the latent space clusters will lead to reduction in the data required for training a good model? Current empirical results show that the difference between performance on the full dataset and the synthesized one is significant. Is it even possible to catch up with the original performance in this way?

2 - How one can guarantee that generated samples indeed will be significantly different from the copyright data, if that one was used in training CLIP mode?

3 - How do you select amount of clusters for generating "prototypes"?

4 - Why not to perform clustering right away in the joint space of representations, but try to map clusters on each other later?

---

> ### Author Response · Authors · 2025-11-20
> **Response to Reviewer WZ3d**
>
> Dear reviewer WZ3d,
>
> Thank you for your valuable feedback and comments. We appreciate your recognition of the need for more compact and easily produced datasets for multimodal training, as well as your positive remarks on our extensive experiments and ablation studies. We will address your questions and concerns in the response below.
>
> **[W1 and Q1] Guarantees and performance of distilled data**
>
> Averaging embeddings within the same cluster produces prototypes that interpolate the diverse semantic features of the original samples. As illustrated in Figure 3 of the manuscript, images generated from these prototypes are semantically enriched. This semantic interpolation also contributes to the performance advantage of PDS over methods that simply select subsets of real data, as reported in Table 2 of the manuscript. Therefore, generating images from these prototypes can reduce the amount of data required for training.
>
> Matching the performance of the full dataset becomes feasible in the extreme case where the number of distilled pairs equals the size of the original dataset. In this scenario, each data point forms its own cluster, and its embedding becomes a prototype. Consequently, the generated data preserves most of the information contained in the original dataset, leading to comparable performance. Experimentally, we observe a monotonic improvement in performance as the number of distilled pairs increases, with the results gradually approaching those obtained from the full dataset for larger distilled dataset sizes. To examine this trend, we distilled Flickr30K using a wide range of distilled dataset sizes and evaluated each distilled set using a ResNet-50 backbone. The table below presents these improvements. We have added this ablation study to Appendix C.7 of the revised manuscript.
> \begin{array}{ccccccc}
>     \hline
> \textbf{Pairs}
> & \textbf{IR@1} & \textbf{IR@5} & \textbf{IR@10}
> & \textbf{TR@1} & \textbf{TR@5} & \textbf{TR@10} \newline
> \hline
> 100  & 7.9 \pm 0.3 & 25.8 \pm 0.4 & 37.3 \pm 0.3 & 10.2 \pm 0.3 & 28.2 \pm 0.9 & 39.0 \pm 0.3 \newline
> 300  & 14.4 \pm 0.4 & 38.1 \pm 0.2 & 51.4 \pm 0.4 & 18.7 \pm 0.5 & 45.0 \pm 0.4 & 57.8 \pm 0.6 \newline
> 500  & 17.1 \pm 0.2 & 43.0 \pm 0.2 & 56.4 \pm 0.1 & 23.9 \pm 0.9 & 50.5 \pm 0.6 & 62.0 \pm 0.3 \newline
> 1000 & 19.5 \pm 0.3 & 46.4 \pm 0.4 & 58.9 \pm 0.2 & 27.4 \pm 0.4 & 55.3 \pm 0.6 & 67.5 \pm 0.6 \newline
> 1500 & 20.5 \pm 0.2 & 48.7 \pm 0.3 & 62.5 \pm 0.2 & 28.8 \pm 0.5 & 59.2 \pm 0.8 & 71.2 \pm 0.4 \newline
> \text{Full Dataset} & 28.5 \pm 0.2 & 59.6 \pm 0.1 & 71.4 \pm 0.1 & 46.0 \pm 0.6 & 76.2 \pm 0.3 & 84.4 \pm 0.2 \newline
> \hline
> \end{array}
>
> **[W2] Motivation clarification regarding pre-trained CLIP**
>
> Although training a large pre-trained model such as CLIP requires a large-scale dataset, this does not contradict the motivation of our method. Our goal is to efficiently distill datasets, including those that were not used to train the pretrained model itself, for scenarios such as neural architecture search, hyperparameter tuning, and continual learning, where models must be trained repeatedly or under constrained data budgets. In such settings, reusing a pre-trained model to distill datasets is especially beneficial.
>
> Existing dataset distillation methods typically require training a model from scratch multiple times for each dataset and using the model parameters obtained during training to optimize the distilled set, which makes the overall procedure computationally expensive. In contrast, our method uses the pre-trained model to perform the distillation without any additional training. Therefore, although CLIP itself was trained on large-scale data, our approach significantly reduces the computational cost of distilling the dataset compared to existing methods, which is the core motivation of our work.
>
> **[W3 and Q2] Copyright-safety clarification**
>
> As you correctly noted, images generated by a generative model do not inherently avoid copyright concerns, and copyrighted training data may still be reflected in the generated samples. Our intention was not to suggest that our approach fully resolves copyright issues. Rather, we aimed to emphasize that PDS synthesizes images using a generative model conditioned on prototypes obtained through interpolation of diverse features. This contrasts with existing methods that add architecture-specific adversarial perturbations to real images, which often produce distilled pairs that are nearly identical to the original samples.
>
> We acknowledge that this intention was not sufficiently clear in the current manuscript. In the revised manuscript, we have clarified this point in the introduction (page 2) and revised the ablation studies (page 8), removing statements that could be interpreted as addressing copyright-related concerns.

---

> ### Author Response · Authors · 2025-11-20
> **Response to Reviewer WZ3d**
>
> **[Q3] Selection of the number of prototypes**
>
> We set the number of clusters to match the distilled dataset sizes commonly used in prior distillation work. In practice, the number of clusters can be chosen heuristically based on the structure of the data in the embedding space. When the embeddings form well-separated and compact groups, a smaller number of clusters is usually sufficient, whereas more diffuse or spread out embeddings often benefit from using a larger number of clusters.
>
> **[Q4] Why not cluster directly in the joint space?**
>
> It is certainly possible to perform clustering directly in a joint embedding space, but the two modalities can differ in how their embeddings are distributed, and this difference can sometimes cause the clustering to be influenced more by one modality than the other. For this reason, we cluster each modality separately and then match the resulting clusters across modalities.
>
> To compare the joint clustering approach (Joint) with the separate clustering approach (Separate), we distilled the Flickr30K and MS-COCO datasets into 100 and 300 samples, and we evaluated all distilled sets using a ResNet-50 backbone. As shown in the table below, the two approaches perform very similarly, and the separate clustering approach is slightly better in a few cases. We have added this ablation study to Appendix C.12 of the revised manuscript.
> \begin{array}{cclccccccc}
> \hline
> \textbf{Dataset} & \textbf{Pairs} & \textbf{Methods} &
> \textbf{IR@1} & \textbf{IR@5} & \textbf{IR@10} &
> \textbf{TR@1} & \textbf{TR@5} & \textbf{TR@10} \newline
> \hline
> \text{Flickr30K} & 100 & \text{Joint} & 8.3 \pm 0.1 & 25.3 \pm 0.2 & 36.8 \pm 0.4 & 11.0 \pm 0.4 & 29.5 \pm 0.4 & 40.7 \pm 0.4 \newline
>  & 100& \text{Separate} & 7.9 \pm 0.3 & 25.8 \pm 0.4 & 37.3 \pm 0.3 &
> 10.2 \pm 0.3 & 28.2 \pm 0.9 & 39.0 \pm 0.3 \newline
> & 300 & \text{Joint} & 14.6 \pm 0.5 & 38.8 \pm 0.3 & 52.3 \pm 0.4 &
> 17.5 \pm 0.4 & 43.5 \pm 0.4 & 56.4 \pm 0.8 \newline
>  & 300 & \text{Separate} & 14.4 \pm 0.4 & 38.1 \pm 0.2 & 51.4 \pm 0.4 &
> 18.7 \pm 0.5 & 45.0 \pm 0.4 & 57.8 \pm 0.6 \newline
> \hline
> \text{MS-COCO} & 100 & \text{Joint} & 2.5 \pm 0.1 & 10.2 \pm 0.2 & 17.5 \pm 0.3 &
> 4.0 \pm 0.1 & 12.9 \pm 0.2 & 20.6 \pm 0.2 \newline
> & 100& \text{Separate} & 2.8 \pm 0.1 & 10.0 \pm 0.2 & 17.3 \pm 0.3 &
> 4.5 \pm 0.2 & 14.0 \pm 0.3 & 21.4 \pm 0.4 \newline
> & 300 & \text{Joint} & 4.9 \pm 0.1 & 16.6 \pm 0.2 & 25.9 \pm 0.3 &
> 6.6 \pm 0.2 & 19.3 \pm 0.3 & 29.2 \pm 0.4 \newline
> & 300& \text{Separate} & 5.3 \pm 0.2 & 17.2 \pm 0.4 & 27.2 \pm 0.6 &
> 7.4 \pm 0.3 & 20.7 \pm 0.3 & 30.2 \pm 0.4 \newline
> \hline
> \end{array}
>
> **Minor comment**
>
> In the revised manuscript, we have provided more detailed explanations of the figures and tables referenced in Section 3, clarifying what each one shows and how it supports our claims.
>
> Please let us know if you have any further concerns.
>
> Sincerely,
> Authors

---

### Official Review · Reviewer_1wEY · 2025-11-07

**Soundness:** 3
**Presentation:** 3
**Contribution:** 3
**Rating:** 6
**Confidence:** 3

**Summary:**

The paper proposes the Prototype-Guided Data Synthesis (PDS) framework, a learning-free dataset distillation framework that eliminates large-scale training and optimization based techniques. The framework uses a pre-trained CLIP multimodal model to obtain image and text embeddings, the embeddings are pruned based on a similarity score and clustered to capture the dataset's semantic diversity through a Mini-Batch K-Means algorithm and matched using the Hungarian algorithm. An unCLIP decoder model is then used on the CLIP image embeddings to generate a synthetic image by retrieving the most similar caption most similar to the text prototype from the training set. The PDS framework outperforms multimodal dataset distillation baseline such as TESLA-VL and LoRS, also the framework outperforms dataset subset selection method such as Herding.

**Strengths:**

The paper proposes a novel dataset distillation method based on a pre-trained CLIP encoder and unCLIP decoder to extract image embeddings. These extract embeddings are then forwarded in an unCLIP decoder to generate a distilled dataset. The paper's methodological presentation and its contributions are well articulated in the text. Empirically, PDS achieves state-of-the-art performance compared with dataset subset selection and multimodal dataset distillation baselines, demonstrating the advantages of the proposed approach.

**Weaknesses:**

- The paper's presented  PDS framework is evaluated only with ViT-L/14 CLIP encoders.
- The code to replicate results is not yet released (even anonymously).
- Which unCLIP decoder is used in the PDS framework? Currently, the authors provide a citation to Ho & Salimans (2022) and the guidance scale and sampling step hyperparameters, without specifying the model architecture used.

**Questions:**

- Can the PDS framework improvements over state-of-the-art methods be reproduced by using different CLIP variants? For example, clip-ViT-B-32 or clip-ViT-B-16?
- Are the authors planning to release the code to reproduce the results reported in the paper? Please include scripts to reproduce all tables/figures.
- Which unCLIP decoder is used? How sensitive are the results to the decoder's guidance scale and sampling step hyperparameters?
- Could the sensitivity of PDS to the similarity-pruning threshold, the clustering seed, and the number of desired distilled samples (M) be assessed, with ablation studies on these factors?

---

> ### Author Response · Authors · 2025-11-20
> **Response to Reviewer 1wEY**
>
> Dear reviewer 1wEY,
>
> Thank you for your valuable feedback and comments. We appreciate your recognition of our novel method, positive remarks on the clarity of our presentation and contributions, and acknowledgment of the strong empirical performance of PDS. We will address your questions and concerns in the response below.
>
> **[W1 and Q1] Performance across different CLIP variants**
>
> There are two publicly available pre-trained unCLIP models:
> - stabilityai/stable-diffusion-2-1-unclip-small, trained with a ViT-L/14 CLIP encoder
> - stabilityai/stable-diffusion-2-1-unclip, trained with a ViT-H/14 CLIP encoder.
>
> For our main experiments, we adopted the ViT-L/14 CLIP model because it is more computationally efficient for extracting image and text features.
>
> We additionally evaluated PDS using the ViT-H/14 CLIP model to examine whether the improvement generalizes to a different CLIP backbone. As shown in the table below, using the ViT-H/14 encoder results in improved performance, likely due to its stronger alignment between image and text features. To obtain these results, we distilled Flickr30K into 300 samples and evaluated the distilled data using both ResNet-50 and ViT backbones. Importantly, our PDS framework continues to outperforms state-of-the-art baselines even with this CLIP variant, indicating that the observed gains are not specific to ViT-L/14. We have added this experiment to Appendix C.8 of the revised manuscript.
> \begin{array}{clcccccc}
> \hline
> \textbf{Backbone} &
> \textbf{CLIP} &
> \textbf{IR@1} & \textbf{IR@5} & \textbf{IR@10} &
> \textbf{TR@1} & \textbf{TR@5} & \textbf{TR@10} \newline
> \hline
> \text{ResNet} & \text{ViT-L/14}
> & 14.4 \pm 0.4 & 38.1 \pm 0.2 & 51.4 \pm 0.4 & 18.7 \pm 0.5 & 45.0 \pm 0.4 & 57.8 \pm 0.6 \newline
>  & \text{ViT-H/14}
> & 17.2 \pm 0.5 & 42.6 \pm 0.3 & 55.6 \pm 0.4 & 23.7 \pm 0.9 & 47.3 \pm 0.8 & 60.0 \pm 0.8 \newline
> \hline
> \text{ViT} & \text{ViT-L/14}
> & 9.1 \pm 0.1 & 27.3 \pm 0.4 & 38.4 \pm 0.4 & 9.6 \pm 0.3 & 26.1 \pm 0.5 & 37.5 \pm 1.2 \newline
> & \text{ViT-H/14}
> & 12.0 \pm 0.1 & 32.8 \pm 0.3 & 43.2 \pm 0.6 & 12.7 \pm 0.6 & 32.4 \pm 0.2 & 44.3 \pm 0.2 \newline
> \hline
> \end{array}
>
> **[W2 and Q2] Code release**
>
> We would like to clarify that all code required to reproduce our results has already been provided in the supplementary material. The accompanying README file provides detailed instructions and scripts for reproducing the results. After acceptance, we will make the full implementation publicly available (e.g., via GitHub).

---

> ### Author Response · Authors · 2025-11-20
> **Response to Reviewer 1wEY**
>
> **[W3, Q3, and Q4] unCLIP decoder details and hyperparameter sensitivity**
>
> We use the publicly available pre-trained unCLIP model stabilityai/stable-diffusion-2-1-unclip-small, which is trained with a ViT-L/14 CLIP encoder.
>
> We find that PDS is overall robust to both the decoder hyperparameters and the PDS-specific hyperparameters. To analyze these factors, we distilled Flickr30K and evaluated the resulting distilled pairs using a ResNet-50 backbone.
>
> We first examined the sensitivity to the decoder hyperparameters. Varying the guidance scale $\in \\{3, 5, 7, 10, 15\\}$ and the number of sampling steps $\in \\{25, 50, 100, 150, 200\\}$ yielded consistent retrieval accuracy, with only slight degradation at very low values where the prototype conditioning becomes weak.
> These trends are shown in Figure 6 in the appendix, which reports the aggregated retrieval accuracy, computed as the average of IR@1, IR@5, IR@10, TR@1, TR@5, and TR@10, using distilled sets of 100 and 300 pairs.
>
> We then analyzed the sensitivity to the PDS-specific hyperparameters. Performance remained consistent across different clustering seeds, as illustrated by the tight box plot in Figure 6 in the appendix, using 20 different seeds. The same figure also presents the effect of the similarity-pruning threshold ratio $\in\\{0.0, 0.1, 0.2, 0.3, 0.5\\}$, and we observe that moderate pruning reliably improves accuracy by removing weakly correlated samples. In contrast, the number of distilled pairs $M$ has a more noticeable impact, with larger distilled sets yielding higher performance, as summarized in the table below. We have added this extensive ablation study to Appendix C.6 and C.7 of the revised manuscript.
> \begin{array}{ccccccc}
>     \hline
> \textbf{Pairs}
> & \textbf{IR@1} & \textbf{IR@5} & \textbf{IR@10}
> & \textbf{TR@1} & \textbf{TR@5} & \textbf{TR@10} \newline
> \hline
> 100  & 7.9 \pm 0.3 & 25.8 \pm 0.4 & 37.3 \pm 0.3 & 10.2 \pm 0.3 & 28.2 \pm 0.9 & 39.0 \pm 0.3 \newline
> 300  & 14.4 \pm 0.4 & 38.1 \pm 0.2 & 51.4 \pm 0.4 & 18.7 \pm 0.5 & 45.0 \pm 0.4 & 57.8 \pm 0.6 \newline
> 500  & 17.1 \pm 0.2 & 43.0 \pm 0.2 & 56.4 \pm 0.1 & 23.9 \pm 0.9 & 50.5 \pm 0.6 & 62.0 \pm 0.3 \newline
> 1000 & 19.5 \pm 0.3 & 46.4 \pm 0.4 & 58.9 \pm 0.2 & 27.4 \pm 0.4 & 55.3 \pm 0.6 & 67.5 \pm 0.6 \newline
> 1500 & 20.5 \pm 0.2 & 48.7 \pm 0.3 & 62.5 \pm 0.2 & 28.8 \pm 0.5 & 59.2 \pm 0.8 & 71.2 \pm 0.4 \newline
> \text{Full Dataset} & 28.5 \pm 0.2 & 59.6 \pm 0.1 & 71.4 \pm 0.1 & 46.0 \pm 0.6 & 76.2 \pm 0.3 & 84.4 \pm 0.2 \newline
> \hline
> \end{array}
>
> Please let us know if you have any further concerns.
>
> Sincerely,
> Authors

---

### Official Review · Reviewer_Td75 · 2025-11-12

**Soundness:** 3
**Presentation:** 2
**Contribution:** 3
**Rating:** 6
**Confidence:** 4

**Summary:**

The aim of this paper is to summarize big multimodal datasets in smaller sizes datasets of hundreds of pairs while losing as little performance as possible. This idea of dataset distillation is old, but has not been explored for multimodal datasets made of pairs. Indeed, these datasets are used as a bridge between two domains and the idea is to make this bridge very light and portable.

**Strengths:**

- The core concept is both timely and important. The paper correctly identifies that summarizing a link between domains is a different and more nuanced problem than summarizing a single domain. This is a valuable contribution to the field.
- The procedure described is logical and well-conceived. The idea of selecting only the overlapping pairs within matched clusters to form prototypes is a particularly clever mechanism for strengthening cross-modal alignment.

**Weaknesses:**

- A thing I found less convincing is what to do with these distilled, let’s say, 300 pairs. From what I understood it is not possible to use them to train a CLIP-like model from scratch (it would have been cool…).
- The application suggested—to use these pairs to link a vision space with a text space through a fine-tuned linear layer—is weaker than it seems. The two spaces are often already very aligned (see e.g., e.g., Huh et al., 2024, "The Platonic Representation Hypothesis"), so it's reasonable to expect they do not need much to be paired. Consequently, the authors should compare their method not only to other distillation techniques but also to methods specifically for multimodal latent space alignment, such as the relative representation alignment in ASIF (Norelli et al., 2023), that already uses far less pairs than usual multimodal datasets.

A cool application of the method I see is to make techniques like ASIF work with even less data, that is: hundreds of pairs. That would be very cool; the reduced dataset would also foster interpretability besides speed and privacy.

- The comparison with other data distillation techniques does not seem entirely fair, since this method saves a text embedding and not a text for each pair, which is significantly more information. In this sense, it would have been good to show performance also of the method selecting the closest caption to the embedding, especially since the authors already retrieve this exact caption to condition the unCLIP image generator.

**Questions:**

- Just a curiosity, have you tried to use unCLIP on the text embedding used in place of the image embedding? I would bet that performance is worse, but maybe not, I am not confident. Perhaps an average of the two is even better?
- One thing that surprised me is that you had clusters with no overlapping pairs at all. How many clusters? Also, I was expecting you to discard them since you cannot do the average, but instead you keep the centroid. Perhaps you want to motivate this choice more, since my first thought as a reader was that you would have discarded them.

---

Overall, I think the topic is important and the idea is interesting, but you could make it better shine! I will set my score at 6 with the hope of raising it.

---

> ### Author Response · Authors · 2025-11-20
> **Response to Reviewer Td75**
>
> Dear reviewer Td75,
>
> Thank you for your valuable feedback and comments. We appreciate your recognition of the importance of our core concept and the unique challenge of summarizing cross-domain links, as well as your positive remarks on the overall mechanism of our proposed method. We are also grateful for your suggestions on potential applications. We will address your questions and concerns in the response below.
>
> **[W1] Practical applications of distilled datasets**
>
> As noted in the introduction, compact distilled datasets enable faster training, which facilitates rapid benchmarking of models and training strategies, making them particularly useful for tasks such as neural architecture search and hyperparameter tuning. In addition, distilled datasets are beneficial in continual learning settings, where a model needs to quickly adapt to new data or tasks. We have revised the introduction to more clearly present these practical advantages on page 1 of the revised manuscript.
>
> **[W2] Application to ASIF**
>
> We appreciate the reviewer’s insight that our distilled data could be particularly useful for methods like ASIF [1]. We agree that this is an appealing application, and our results indeed confirm that PDS can allow ASIF to operate effectively with fewer pairs, potentially extending the utility of data distillation beyond compression.
>
> Although ASIF and PDS were developed for different purposes, they are complementary. ASIF aligns pre-trained unimodal encoders without additional training by computing relative representations whose elements are cosine similarities between a sample and each anchor in the anchor set.
> Consequently, the quality of the anchor set is crucial, and the computational cost of ASIF increases with the number of anchors. The original ASIF paper reports that more than one million anchors are required to reach CLIP-level performance, highlighting the importance of identifying compact yet informative anchors.
>
> PDS can address this point by producing a small set of highly informative samples that can serve as stronger anchors than those obtained by selecting subsets of the original dataset. In Table 2 of the manuscript, we already show that PDS produces more representative samples than subset selection baselines. Building on this result, we evaluated the performance of ASIF with 300 anchors obtained from Flickr30K by (i) subset selection methods, (ii) other distillation methods, and (iii) PDS.
> For this comparison, we used a ResNet-50 image encoder and a CLIP text encoder to compute the relative representations, and performance was evaluated through retrieval accuracy on Flickr30K. As shown in the table below, the anchors produced by PDS led to the strongest performance among all compared methods.
>
> These findings support the reviewer’s intuition that methods such as ASIF can benefit from distilled data, since PDS can provide a compact yet informative anchor set that outperforms anchors obtained through subset selection or other distillation methods. We have added this application to Appendix D.1 of the revised manuscript.
> \begin{array}{lcccccc}
> \hline
> \textbf{Methods}
> & \textbf{IR@1} & \textbf{IR@5} & \textbf{IR@10}
> & \textbf{TR@1} & \textbf{TR@5} & \textbf{TR@10} \newline
> \hline
> \text{K-center} & 3.8 & 14.4 & 24.2 & 4.0 & 14.8 & 22.7 \newline
> \text{Herding}  & 6.0 & 21.5 & 32.5 & 6.9 & 19.1 & 28.1 \newline
> \text{CLIP score} & 2.1 & 9.5  & 16.9 & 3.5 & 11.5 & 18.7 \newline
> \text{LAION filtering} & 3.1 & 11.2 & 17.9 & 3.4 & 12.5 & 17.4 \newline
> \text{Image-based}  & 2.3 & 9.6  & 17.1 & 4.1 & 12.6 & 18.3 \newline
> \text{TESLA-VL}  & 5.3 & 19.4 & 29.8 & 5.0 & 16.8 & 25.9 \newline
> \text{LoRS}   & 4.9 & 18.6 & 28.2 & 5.7 & 16.6 & 25.9 \newline
> \textbf{PDS}  & \textbf{9.9} & \textbf{29.8} & \textbf{41.8} & \textbf{9.6} & \textbf{27.7} & \textbf{38.6} \newline
> \hline
> \end{array}
> [1] Norelli et al. ASIF: Coupled data turns unimodal models to multimodal without training. NeurIPS 2023.
>
> **[W3] Fairness of comparison with other distillation methods**
>
> We believe that the comparison between PDS and other dataset distillation approaches is fair.
> Since discrete text cannot be directly optimized, all existing methods optimize continuous text representations and use them as the distilled dataset. Therefore, our use of text embeddings aligns with the procedure commonly adopted in other distillation methods.

---

> ### Author Response · Authors · 2025-11-20
> **Response to Reviewer Td75**
>
> **[Q1] unCLIP decoder conditioning with text or averaged embeddings**
>
> We conducted additional experiments in which the unCLIP decoder was conditioned either on the text prototype alone or on the average of the text and image prototypes. As shown in the table below, conditioning on the text prototype alone resulted in degraded performance, while averaging the text and image prototypes produced results comparable to our original approach with no noticeable improvements. To perform this comparison, we distilled Flickr30K into 100 samples and conducted the evaluation using the ResNet-50 and ViT backbones. We have added this ablation study in Appendix C.11 of the revised manuscript.
> \begin{array}{llcccccc}
> \hline
> \textbf{Backbone} & \textbf{Methods}
> & \textbf{IR@1} & \textbf{IR@5} & \textbf{IR@10}
> & \textbf{TR@1} & \textbf{TR@5} & \textbf{TR@10} \newline
> \hline
> \text{ResNet}
> & \text{Text} & 8.1 \pm 0.2 & 24.9 \pm 0.2 & 36.6 \pm 0.4 & 10.8 \pm 0.4 & 28.2 \pm 0.7 & 38.3 \pm 0.7 \newline
> & \text{Average} & 7.9 \pm 0.2 & 26.2 \pm 0.5 & 37.9 \pm 0.5 & 11.1 \pm 0.3 & 28.2 \pm 0.5 & 39.4 \pm 0.3 \newline
> & \text{Image} & 7.9 \pm 0.3 & 25.8 \pm 0.4 & 37.3 \pm 0.3 & 10.2 \pm 0.3 & 28.2 \pm 0.9 & 39.0 \pm 0.3 \newline
> \hline
> \text{ViT}
> & \text{Text} & 5.6 \pm 0.2 & 17.3 \pm 0.2 & 26.4 \pm 0.1 & 4.6 \pm 0.4 & 16.3 \pm 0.7 & 23.9 \pm 0.6 \newline
> & \text{Average} & 5.9 \pm 0.1 & 18.5 \pm 0.1 & 27.7 \pm 0.3 & 6.5 \pm 0.6 & 18.4 \pm 0.3 & 27.3 \pm 0.3 \newline
> & \text{Image}  & 6.8 \pm 0.3 & 19.2 \pm 0.3 & 28.5 \pm 0.4 & 6.6 \pm 0.5 & 17.5 \pm 0.5 & 26.9 \pm 0.5 \newline
> \hline
> \end{array}
>
> **[Q2] Analysis of matched clusters without overlapping pairs**
>
> In short, the number of matched clusters without overlapping pairs (non-overlapping clusters) is extremely small when the dataset is distilled into 100 or 300 samples, and whether we keep or discard them yields nearly identical performance. For this reason, we keep their centroids to maintain the intended distilled dataset size.
>
> To more comprehensively assess the effect of such non-overlapping clusters, we applied the two approaches (Keeping and Discarding) across a wide range of distilled dataset sizes. We conducted experiments on Flickr30K by distilling it into various numbers of samples and evaluating the resulting distilled datasets using a ResNet-50 backbone. In small-scale settings, both approaches show nearly identical performance because only a few non-overlapping clusters arise. However, as the distilled size increases, the number of such clusters grows noticeably, as shown in the table below.
> \begin{array}{lccccc}
> \hline
> \text{\textbf{Pairs}} & \textbf{100} & \textbf{300} & \textbf{500} & \textbf{1000} & \textbf{1500} \newline
> \hline
> \textbf{Number of non-overlapping clusters}
> & 1 & 8 & 24 & 87 & 166 \newline
> \hline
> \end{array}
> These non-overlapping clusters tend to have misaligned centroids (see Figure 5 in the appendix), which weakens cross-modal alignment and leads to the performance degradation reported in the table below. In these larger-scale regimes, discarding non-overlapping clusters is therefore preferable. We have added this clarification to the main text (page 5) and the corresponding details to Appendix C.3 of the revised manuscript.
> \begin{array}{llcccccc}
> \hline
> \textbf{Pairs} & \textbf{Methods}
> & \textbf{IR@1} & \textbf{IR@5} & \textbf{IR@10}
> & \textbf{TR@1} & \textbf{TR@5} & \textbf{TR@10} \newline
> \hline
> 100 & \text{Keeping}
> & 7.9 \pm 0.3 & 25.8 \pm 0.4 & 37.3 \pm 0.3 & 10.2 \pm 0.3 & 28.2 \pm 0.9 & 39.0 \pm 0.3 \newline
> & \text{Discarding}
> & 7.8 \pm 0.2 & 25.9 \pm 0.4 & 37.5 \pm 0.2 & 10.0 \pm 0.3 & 27.5 \pm 1.2 & 38.8 \pm 0.9 \newline
> \hline
> 300 & \text{Keeping}
> & 14.4 \pm 0.4 & 38.1 \pm 0.2 & 51.4 \pm 0.4 & 18.7 \pm 0.5 & 45.0 \pm 0.4 & 57.8 \pm 0.6 \newline
> & \text{Discarding}
> & 14.9 \pm 0.2 & 38.6 \pm 0.5 & 51.9 \pm 0.3 & 19.3 \pm 0.4 & 45.0 \pm 0.5 & 57.2 \pm 0.6 \newline
> \hline
> 500 & \text{Keeping}
> & 16.3 \pm 0.2 & 41.3 \pm 0.2 & 54.9 \pm 0.4 & 22.5 \pm 0.3 & 49.3 \pm 0.8 & 61.4 \pm 0.4 \newline
> & \text{Discarding}
> & 17.1 \pm 0.2 & 43.0 \pm 0.2 & 56.4 \pm 0.1 & 23.9 \pm 0.9 & 50.5 \pm 0.6 & 62.0 \pm 0.3 \newline
> \hline
> 1000 & \text{Keeping}
> & 18.2 \pm 0.3 & 43.5 \pm 0.2 & 56.4 \pm 0.5 & 24.3 \pm 0.4 & 52.7 \pm 0.6 & 64.7 \pm 0.6 \newline
> & \text{Discarding}
> & 19.5 \pm 0.3 & 46.4 \pm 0.4 & 58.9 \pm 0.2 & 27.4 \pm 0.4 & 55.3 \pm 0.6 & 67.5 \pm 0.6 \newline
> \hline
> 1500 & \text{Keeping}
> & 18.9 \pm 0.2 & 45.6 \pm 0.2 & 59.4 \pm 0.3 & 26.5 \pm 0.3 & 55.5 \pm 0.8 & 67.8 \pm 1.2 \newline
> & \text{Discarding}
> & 20.5 \pm 0.2 & 48.7 \pm 0.3 & 62.5 \pm 0.2 & 28.8 \pm 0.5 & 59.2 \pm 0.8 & 71.2 \pm 0.4 \newline
> \hline
> \end{array}
>
> Please let us know if you have any further concerns.
>
> Sincerely,
> Authors

---

### Author Response · Authors · 2025-12-03
**Summary of the Rebuttal Discussions**

Dear AC,

We sincerely appreciate your efforts in evaluating our submission during this challenging review cycle.

Below is a quick summary of our review and rebuttal discussions.

The reviewers positively acknowledged that our work addresses an important problem (Td75, WZ3d), provides clear and well-articulated methodological contributions (1wEY, Td75), and introduces a novel and practical learning-free multimodal distillation framework with strong cross-architecture generalization (1wEY, x6Kp, gG1N), supported by extensive experiments and ablation studies, along with its demonstrated efficiency (WZ3d, gG1N).

The reviewers raised several concerns, noting that adequately resolving them could positively influence their overall assessment.
Their main points focused on clarification regarding the practical applications of the distilled dataset (Td75), the method’s sensitivity and robustness (1wEY, WZ3d, x6Kp, gG1N), its generality across datasets and CLIP variants (1wEY, gG1N), and the rationale behind several design choices (Td75, WZ3d).

In the rebuttal, we addressed these concerns by:

- demonstrating the practical utility of the distilled dataset through its application to ASIF (Appx. D.1);

- providing detailed analyses of hyperparameter sensitivity and scaling behavior (Appx. C.6, C.7);

- evaluating generality with an additional dataset and CLIP backbone, and robustness on rare classes (Appx. C.8, C.9, C.10);

- clarifying design choices such as the decoder conditioning and clustering strategy (Appx. C.3, C.11, C.12).

We believe that our rebuttal has successfully addressed all the reviewers' concerns.

Thank you very much for your time and consideration.

Best regards,

Authors

---

### Meta-Review · Area_Chair_KL2v · 2026-01-07

**Summary:**

The paper proposes a learning-free framework for multimodal dataset distillation, and the reviewers broadly agree that this is a timely and practically relevant direction. The approach is technically sound, clearly motivated, and empirically competitive, particularly in settings where only very small distilled sets are feasible and cross-architecture generalization is required.

The discussion and rebuttal improve the paper’s clarity and strengthen the empirical basis of the claims, especially with respect to robustness, generalization, and practical usage. The authors also appropriately acknowledge the method’s dependencies and limitations, and position the contribution more carefully relative to pretrained models and existing distillation approaches.

Overall, the work offers a coherent and useful contribution rather than a purely incremental one. The remaining concerns mainly relate to scope, positioning, and reliance on existing foundation models, and do not undermine the validity or relevance of the proposed method. I therefore recommend acceptance.

**Reviewer Concerns:**

In the initial reviews, four reviewers give scores around the acceptance threshold (Td75, 1wEY, WZ3d, x6Kp), while Reviewer gG1N gives a below-threshold score.

The borderline reviewers’ concerns concentrate on (i) experimental completeness and robustness (e.g., hyperparameter sensitivity, CLIP variants, and evaluation beyond the two main datasets), (ii) implementation specificity and reproducibility (unCLIP decoder details and code/scripts), and (iii) the practical use and fairness of the evaluation protocol (e.g., the use of text embeddings). In the rebuttal, the authors add targeted experiments (e.g., sensitivity/scaling analyses), clarify the decoder choice and reproducibility via the supplementary material, and include an ASIF application. I therefore expect these reviewers’ scores to remain stable or increase slightly.

Reviewer gG1N’s concerns center on potential representation degradation from prototype averaging, dependence on the unCLIP decoder and possible failure cases, generality beyond Flickr30K/MS-COCO, and behavior on rare/long-tail samples. The response adds additional evidence (extra dataset and rare-sample evaluation) and makes the limitations and failure regimes explicit. Given this, I expect the reviewer’s score to move closer to the borderline/accept range.

**Reviewer Scores:**

See above

---

### Decision · Program_Chairs · 2026-01-26

Accept (Poster)